# PIED: PHYSICS-INFORMED EXPERIMENTAL DESIGN FOR INVERSE PROBLEMS

**Apivich Hemachandra**\*†**, Gregory Kang Ruey Lau**\*†‡**,**
†Department of Computer Science, National University of Singapore, Singapore 117417
‡CNRS@CREATE, 1 Create Way, #08-01 Create Tower, Singapore 138602
{apivich,greglau}@comp.nus.edu.sg

**See-Kiong Ng & Bryan Kian Hsiang Low**
Department of Computer Science, National University of Singapore, Singapore 117417
seekiong@nus.edu.sg, lowkh@comp.nus.edu.sg

## ABSTRACT

In many science and engineering settings, system dynamics are characterized by governing partial differential equations (PDEs), and a major challenge is to solve inverse problems (IPs) where unknown PDE parameters are inferred based on observational data gathered under limited budget. Due to the high costs of setting up and running experiments, experimental design (ED) is often done with the help of PDE simulations to optimize for the most informative design parameters (e.g., sensor placements) to solve such IPs, prior to actual data collection. This process of optimizing design parameters is especially critical when the budget and other practical constraints make it infeasible to adjust the design parameters between trials during the experiments. However, existing ED methods tend to require sequential and frequent design parameter adjustments between trials. Furthermore, they also have significant computational bottlenecks due to the need for complex numerical simulations for PDEs, and do not exploit the advantages provided by physics informed neural networks (PINNs) in solving IPs for PDE-governed systems, such as its meshless solutions, differentiability, and amortized training. This work presents Physics-Informed Experimental Design (PIED), the first ED framework that makes use of PINNs in a fully differentiable architecture to perform continuous optimization of design parameters for IPs for one-shot deployments. PIED overcomes existing methods' computational bottlenecks through parallelized computation and meta-learning of PINN parameter initialization, and proposes novel methods to effectively take into account PINN training dynamics in optimizing the ED parameters. Through experiments based on noisy simulated data and even real world experimental data, we empirically show that given limited observation budget, PIED significantly outperforms existing ED methods in solving IPs, including for challenging settings where the PDE parameters are unknown functions rather than just finite-dimensional.

## 1 INTRODUCTION

The dynamics of many systems studied in science and engineering can be described via *partial differential equations* (PDEs). Given the PDE governing the system and the properties of the system (which we refer to as PDE parameters), we can perform *forward simulations* to predict how the system behaves. However, in practice, the true PDE parameters are often unknown, and we are instead interested in recovering the unknown PDE parameters based on observations of the system's behaviors. This problem of recovering the unknown PDE parameters is a type of *inverse problem* (IP) (Vogel, 2002; Ghattas & Willcox, 2021), and have been studied in classical mechanics (Tanaka & Bui, 1993; Gazzola et al., 2018), quantum mechanics (Chadan et al., 1989) or geophysics (Smith et al., 2021; Waheed et al., 2021), and more. It is challenging to directly solve IPs given observational data

---
\*Equal contribution.

since PDE parameters can often affect the behavior of the PDE solution in complex ways. This is further complicated in practice where data acquisition (e.g., making measurements from experiments or field trials) is often costly and so only limited number of observations can be made, making the choice of *which observations to make given a limited budget* also critical to solving IPs.

Experimental design (ED) methods aim to tackle the data scarcity problem by optimizing design parameters, such as sensor placement locations, to yield the most informative measurements for estimating the unknown inverse parameters (Razavy, 2020; Alexanderian, 2021). These methods typically make use of forward simulations based on guesses of the true inverse parameter to find the best design parameter for the measurements. However, these methods are less practical in many PDE-informed IPs due to significant computational bottlenecks in forward simulations and solving IPs, especially for systems with complex forward models and PDEs. In particular, simulators using conventional PDE solvers often are costly to run, and return discretized (mesh-based) approximate solutions which are often incompatible with efficient continuous gradient-based optimization methods.

Physics-Informed Neural Networks (PINNs) are neural networks that incorporate PDEs and their initial/boundary conditions (IC/BCs) into the NN loss function (Raissi et al., 2019), and have been successfully applied to various science problems (Chen et al., 2020; Cai et al., 2021; Jagtap et al., 2022). PINNs are especially well-suited to tackle ED for IPs, as they (1) allow easy incorporation of observational data into the inverse problem solver through the training loss function, (2) can be used for both running forward simulations and directly solving IPs with the same model architecture, (3) are continuous and differentiable w.r.t. the function inputs, and (4) have training whose costs can be amortized, e.g., through transfer learning. However, to our knowledge, there has not been an ED framework for IPs that fully utilizes these advantages from PINNs for ED problems.

In this paper, we present Physics-Informed Experimental Design (PIED), the first ED framework that makes use of PINNs in a fully differentiable architecture to perform continuous optimization of design parameters for IPs for one-shot deployment[1]. Our contributions are summarized as follows:

- We propose a novel ED framework that makes use of PINNs as both forward simulators and inverse solvers in a fully differentiable architecture to perform continuous optimization of design parameters for IPs for one-shot deployments (Sec. 3).
- We introduce the use of a learned initial NN parameter which are used for all PINNs in PIED (Sec. 4.1), based on first-order meta-learning methods, allowing for more efficient PINN training over multiple PDE parameters.
- We present various effective ED criteria based on novel techniques for quantifying the training dynamics of PINNs (Sec. 4.2). These proposed criteria are differentiable w.r.t. the design parameters, and therefore can be optimized efficiently via gradient-based methods.
- We empirically demonstrate that PIED is able to outperform other ED methods on inverse problems (Sec. 5), both in the case where the PDE parameters of interest are finite-dimensional and when they are unknown functions.

## 2  BACKGROUND

In this section, we provide a formalism of the experimental design (ED) problem for PDE-based inverse problems (IPs), and physics-informed neural networks (PINNs) which will be used in our proposed framework. We also discuss existing works on ED and PINNs applied to solving IPs.

### 2.1  PROBLEM SETUP

**Inverse problems.**   Consider a system described by a PDE[2] of the form

$$\mathcal{D}[u, \beta](x) = f(x) \quad \forall x \in \mathcal{X} \qquad \text{and} \qquad \mathcal{B}[u, \beta](x') = g(x') \quad \forall x' \in \partial\mathcal{X} \qquad (1)$$

where $u : \mathcal{X} \to \mathbb{R}^{d_{\text{out}}}$ describes the observable function (solution of the PDE) over a coordinate variable $x \in \mathcal{X} \subset \mathbb{R}^{d_{\text{in}}}$ (where time could be a subcomponent), and $\beta \in \mathbb{R}^{d_{\text{inv}}}$ are PDE parameters[3].

---

[1]The code for the project can be found at `https://github.com/apivich-h/pied`.

[2]Examples of PDEs, specifically those in our experiments, can be found in App. D.

[3]For simplicity we assume $\beta$ is finite-dimensional. In our experiments, we demonstrate how our method can also be extended to cases where $\beta$ is a function of $x$.

$\mathcal{D}$ is a PDE operator and $\mathcal{B}$ is an operator for the initial/boundary conditions (IC/BCs) at boundary $\partial\mathcal{X} \subset \mathcal{X}$. Different PDE parameters $\beta$ results in different observable function $u$ that satisfies (1), which for convenience will be denoted by $u_\beta$.

For inverse problems (IPs), the operators $\mathcal{D}$ and $\mathcal{B}$ and functions $f$ and $g$ are known, and the task is to estimate the unknown PDE parameter of interest, $\beta_0$, that cannot be observed directly. Instead, we can only make noisy measurements of the corresponding observable function $u_{\beta_0}$ at $M$ observation inputs[4] $X = \{x_j\}_{j=1}^M \subset \mathcal{X}$ to get observation values $Y = \{u_{\beta_0}(x_j) + \varepsilon_j\}_{j=1}^M$, where we assume Gaussian noise $\varepsilon_j \sim \mathcal{N}(0, \sigma^2)$. In general, $X$ could possibly be constrained to a set of feasible configurations $\mathcal{S} \subseteq \mathcal{X}^M$. To solve the IP, we could use an *inverse solver* that finds the PDE parameter $\hat{\beta}$ that fits best with the observed data $(X, Y)$ where

$$\hat{\beta}(X, Y) \approx \arg\min_\beta \|u_\beta(X) - Y\|^2. \tag{2}$$

**Experimental design.** Unfortunately, the observations $Y$ are typically expensive to obtain due to costly sensors or operations. In many settings, the observation inputs also have to be chosen in a one-shot rather than in a sequential, adaptive manner due to the costs of reinstalling sensors and inability to readjust the design parameters on-the-fly. Hence, the observation input $X$ should be carefully chosen before actual measurements are made. As we *do not* know the true PDE parameter $\beta_0$, a good observation input should maximize the average performance of the inverse solver over its distribution $p(\beta)$. In the *experimental design* (ED) problem, the goal is therefore to find the observation input $X \in \mathcal{S}$ which minimizes

$$L(X) = \mathbb{E}_{\beta, Y \sim p(\beta)p(Y|\beta)}\left[\left\|\hat{\beta}(X, Y) - \beta\right\|^2\right], \tag{3}$$

or the *expected* error of the estimated PDE parameter w.r.t. the possible true PDE parameters.

ED methods can be deployed in the *adaptive* setting, where multiple rounds of observations are allowed, and the design parameter can be adjusted between each rounds based on the observed data. However, there are many practical scenarios where *non-adaptive* ED are desirable. For instance, field scientists who incur significant cost in planning, deployment, and collection of results for each iteration may strongly prefer to perform a single round of sensor placement and measurements as opposed to sequentially making few measurements per round and frequently adjusting sensor placement locations. Another example is when the phenomena being observed occurs in a single time period, and all observations have to be decided beforehand. In these cases, it is more practical to optimize for a single design parameter upfront and collect all observational data for the IP in one-shot. Unlike past works, our work aims to optimize performance for one-shot deployment.

**Physics-informed neural networks.** To reduce the computational costs during the ED and IP solving processes, we will use physics-informed neural networks (PINNs) (Raissi et al., 2019) to simulate PDE solutions and solve IPs. PINNs are neural networks (NNs) $\hat{u}_\theta$ with NN paramaters $\theta$ that approximates the solution $u$ to (1) by minimizing the composite loss[5]

$$\mathcal{L}(\theta, \beta; X, Y) = \underbrace{\frac{\|\hat{u}_\theta(X) - Y\|_2^2}{2|X|}}_{\mathcal{L}_{\mathrm{obs}}(\theta; X, Y)} + \underbrace{\frac{\|\mathcal{D}[\hat{u}_\theta, \beta](X_p) - f(X_p)\|_2^2}{2|X_p|} + \frac{\|\mathcal{B}[\hat{u}_\theta, \beta](X_b) - g(X_b)\|_2^2}{2|X_b|}}_{\mathcal{L}_{\mathrm{PDE}}(\theta, \beta)},$$

$$\tag{4}$$

where $(X, Y)$ are observational data, and $X_p \subset \mathcal{X}$ and $X_b \subset \partial\mathcal{X}$ are collocation points to enforce PDE and IC/BC constraints respectively. PINNs can be used both as forward simulators when $\beta$ is known but no observational data is available ($\mathcal{L}_{\mathrm{obs}}(\theta; X, Y) = 0$), or as inverse solvers when observational data is available but $\beta$ is unknown and is learned jointly with $\theta$ during PINN training.

## 2.2 RELATED WORKS

Existing ED methods in the literature include those that adopts a Bayesian framework (Chaloner & Verdinelli, 1995; Long et al., 2013; Belghazi et al., 2018; Foster et al., 2019) and those that aims to

---

[4]We abuse notations by allowing the set of $X$ to be an argument for functions which takes in $x \in \mathcal{X}$ as well.

[5]For simplicity we consider one IC/BC, however the loss can be generalized to include multiple IC/BCs by adding similar loss terms for each constraint.

construct a policy for choosing the optimal experimental design (Ivanova et al., 2021; Lim et al., 2022). Many of these ED methods utilize forward simulations that can be easily and efficiently queried from, which is often unavailable in PDE-based IPs due to the need to numerically solve complex PDEs. ED and IP solving methods that rely on numerical simulators (Ghattas & Willcox, 2021; Alexanderian, 2021; Alexanderian et al., 2024) are computationally expensive due to repeated forward simulations required during optimization, and are usually restricted to some PDEs. Furthermore, numerical solvers often require input discretization and cannot be easily differentiated, which restricts the applicable optimization techniques.

Meanwhile, PINNs have been extensively used model systems and solve IPs across many scientific domains (Chen et al., 2020; Cai et al., 2021; Waheed et al., 2021; Bandai & Ghezzehei, 2022; Jagtap et al., 2022; Shadab et al., 2023), due to the simplicity in incorporating observational data with existing PDE from the respective domains, and its ability to recover PDE parameters with one round of PINN training. However, these works assume that the observational data to be used for IPs have already been provided, and do not consider the process of observational data selection and how it may affect the IP performance. Further material on related works are in App. C.

## 3 EXPERIMENTAL DESIGN LOOP

In this section, we first discuss how IPs can be solved with PINNs given a set of observational data $(X, Y)$, and how the choice of observation inputs $X$ matters especially when there is limited observation budget. Thereafter, we present our proposed ED framework, PIED, which optimizes for the observation input while fully utilizing the advantages provided by PINNs.

### 3.1 SOLVING INVERSE PROBLEMS WITH PINNS

The process of solving the IP with PINNs is summarized in Fig. 1a. As presented in Sec. 2, the goal is to find the true PDE parameter $\beta^*$ of the system based on (2) by conducting experiments at a limited number of observation input $X$ to obtain observational data $(X, Y)$, where $Y$ are noisy measurements of the observable function $u_{\beta^*}$. This can be directly solved using a PINN-based inverse solver that is trained to minimize the composite loss in (4) to find the inferred $\hat{\beta}^*$ of the system.

However, given limited observation budget, the choice of observation input $X$ becomes important in achieving a good estimate $\hat{\beta}^*$ – some points are more informative than others, and allows the inverse solver to achieve better estimates with lower variance. Intuitively, we could interpret the limited observation as an information bottleneck, where the information on a system's $\beta$ would be compressed from the entire observable function $u_\beta$ into a relatively low-dimensional, noisy representation. The better the choice of $X$, the better we could extract the value of $\beta^*$ via the inverse solver.

### 3.2 COMPONENTS OF THE ED DESIGN LOOP

To optimize for $X$, ED methods rely on several forward simulations of the system for different reference $\beta$ values before collecting any observational data from actual, costly experiments. Multiple possible $X$ would then need to be tested on all these sets of simulations, to assess the inverse solver's performance. However, existing methods that rely on conventional numerical integration methods do not enable efficient parallel computation and differentiability for gradient-based optimization. Instead, we propose to fully utilize the advantages provided by PINNs to tackle the ED problem.

To achieve this, we propose our ED framework PIED, which is visualized in Fig. 1b. We consider $N$ *parallel threads*, each representing different reference $\beta_i$ values. Similar to existing ED methods, the range of $\beta$ values could be informed by domain knowledge. Within each thread $i$, we have three components: (1) a *forward simulator* that returns an observable function $\tilde{u}_{\beta_i}$ which approximates the PDE solution $u_{\beta_i}$, (2) a *observation selector* that generates observational data based on $\tilde{u}_{\beta_i}(X)$ at observation inputs $X$ (consistent across all threads), and (3) an *inverse solver* that takes in the observational data to produce estimates of the PDE parameter $\hat{\beta}_i$. Finally, we require (4) a *criterion* that captures how good the estimates are across all threads on aggregate, and an *input optimization* method to select the single best observation input according to the criterion.

Figure 1: Comparison between observation selection and solving IPs in real life (Fig. 1a), versus the proess as modelled in the PIED framework (Fig. 1b).

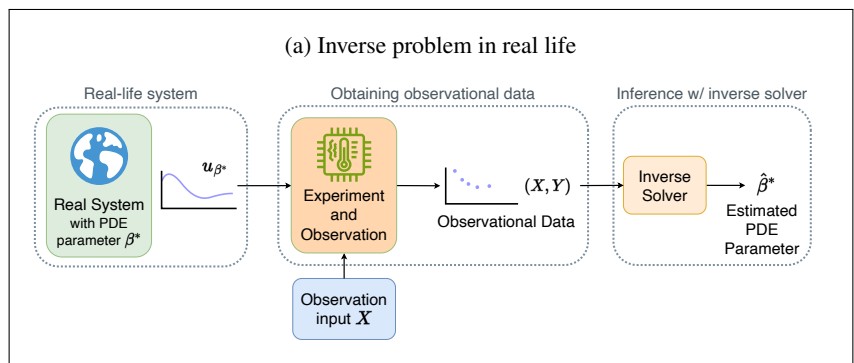

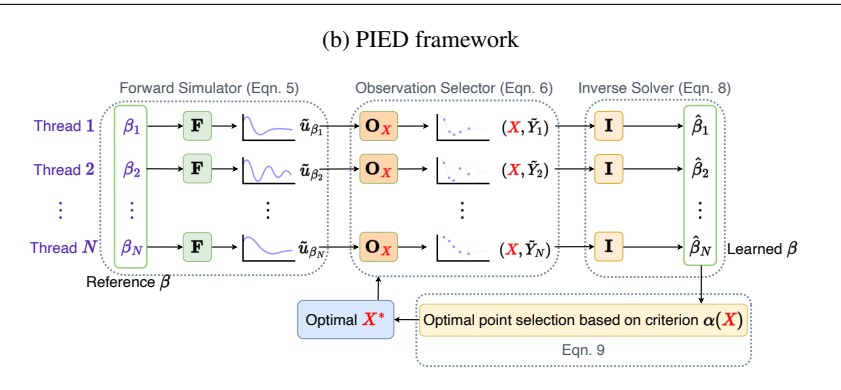

**PINN-based forward simulator (F).**     For the first component, PIED uses PINNs to simulate what the observable function $u_{\beta_i}$ given a reference $\beta_i$ would be in each thread. Specifically, a forward PINN is trained [6] with loss $\mathcal{L}_{\text{PDE}}(\theta, \beta_i)$ from (4) with fixed $\beta_i$ to generate $\tilde{u}_{\beta_i}$ over the input space $\mathcal{X}$,

$$\beta_i \xrightarrow{\quad \text{forward simulator } \mathbf{F} \quad} \tilde{u}_{\beta_i} = \hat{u}_{\theta_i} \approx u_{\beta_i}. \tag{5}$$

Note that unlike classical simulators based on numerical integrators that require fixed discretization of the input space $\mathcal{X}$, the trained PINNs $\tilde{u}_{\beta_i}$ represent learned *functions* that are meshless, and therefore can be queried at and even differentiated w.r.t. any input $x \in \mathcal{X}$. This is a major advantage for ED, as it allows us to flexibly test a continuous range of $X$ without having to re-run the forward simulators many times for different discretization, and also enables efficient gradient-based optimization.

Our use of PINNs as forward simulators also offer significant computational advantages. First, modern software and hardware allows for very efficient training of multiple NNs in parallel (for example, using increasingly accessible and powerful GPUs and via the `vmap` function on JAX), unlike classical simulators. Second, the various PINNs in different thread can all be initialized from a common pre-trained base model (which we discuss in Sec. 4.1), speeding convergence of the PINN during training. This will be even more advantageous if researchers start sharing open-sourced, pre-trained models, similar to what is currently being done for large language models. Third, we can reuse the trained forward PINNs to approximate the dynamics of the inverse PINNs and consequently boost the effectiveness of the downstream ED process, as we will describe in Sec. 4.

**Observation selector ($\mathbf{O}_X$).**     Given the trained forward PINNs $\{\tilde{u}_{\beta_i}\}_{i=1}^N$, the second component applies a "sieve" that queries all forward PINNs with the *same* $M$ observation input $X = \{x_j\}_{j=1}^M$ to produce the respective sets of noisy predicted observations, i.e., for each $i = 1, \dots, N$,

$$\tilde{u}_{\beta_i} \xrightarrow{\quad \text{observation selector } \mathbf{O}_X \quad} (X, \tilde{Y}_i) = \left( \{x_j\}_{j=1}^M, \{\tilde{u}_{\beta_i}(x_j) + \varepsilon_j\}_{j=1}^M \right) \tag{6}$$

---

[6]Note that no actual experimental data is required for the training of forward PINNs, as explained in Sec. 2.

where $\varepsilon_j \sim \mathcal{N}(0, \sigma^2)$ are i.i.d. noise added to model the observational data generation process. The observation input $X$ applied here is the candidate design parameter[7] to be optimized for in the ED problem – different choices of $X$ will yield different sets of observational data $(X, \tilde{Y}_i)$ that impacts the quality of the PDE parameter estimate, as per (3).

**PINN-based inverse solver (I).**   For each $\beta_i$, the predicted observational data $(X, \tilde{Y}_i)$ from (6) will then be used to train an inverse solver PINN (inverse PINN) with loss function $\mathcal{L}(\theta, \beta; X, \tilde{Y}_i)$ from (4) to return an estimated PDE parameter $\hat{\beta}_i$, i.e., for each $i = 1, \ldots, N$,

$$\left(X, \tilde{Y}_i\right) \xrightarrow{\text{inverse solver } \mathbf{I}} \hat{\beta}_i. \tag{7}$$

Note that in our ED framework, both the forward simulators and inverse solvers are PINNs and have the same architecture. This enables us to develop effective approximation techniques based on predicting the dynamics of the inverse PINNs using its corresponding forward PINN, significantly reducing computational cost for the ED process, and also the eventual IP process in Sec. 3.2 that also uses the inverse PINN. We elaborate further on these techniques in Sec. 4.

**Criterion and optimal observation input selection.**   Finally, we could compute the ED criterion in (3) for a given $X$, which evaluates how well the estimates $\hat{\beta}_i$ from the inverse solvers matches the corresponding reference parameter values $\beta_i$ across all parallel threads $i$ on average,

$$\alpha(X) = \frac{1}{N} \sum_{i=1}^{N} \alpha_i(X) = \frac{1}{N} \sum_{i=1}^{N} \left[ -\left\| \hat{\beta}_i(X, \tilde{Y}_i) - \beta_i \right\|^2 \right]. \tag{8}$$

We can then perform optimization using the observation selector and inverse solver components, where we search for the observation input $X$ that maximizes $\alpha(X)$. Note that the same forward PINNs $\{\tilde{u}_{\beta_i}\}_{i=1}^{N}$ could be re-used for all iterations and hence only need to be computed once. As our PINN inverse solver is differentiable, we can back-propagate through $\alpha$ directly to compute $\nabla_X \alpha(X)$. This allows $\alpha$ to be optimized using gradient-based optimization methods, instead of methods such as Bayesian optimization which do not typically perform well on high-dimensional problems, or require combinatorial optimization over discretized observation inputs.

## 4   EFFICIENT IMPLEMENTATION OF PIED

In this section, we introduce training and approximation methods which further boosts the efficiency of PIED. We first propose a meta-learning approach to learn a PINN initialization for efficient fine-tuning of all our forward and inverse PINNs across different threads (Sec. 4.1), before proposing approximations of our criterion in (8) to effectively optimize for the observation inputs (Sec. 4.2).

### 4.1   SHARED META-LEARNED INITIALIZATION FOR ALL PINN-BASED COMPONENTS

In PIED, we could significantly reduce computational time and benefit from amortized training by pre-training and using a shared initialization for all PINN components in our framework. We can efficiently achieve this with REPTILE (Nichol et al., 2018; Liu et al., 2021), a first-order meta-learning algorithm. To see this, note that all PINNs (forward and inverse) in PIED are modelling the same system and may benefit from joint training, but have different PDE parameters $\beta_i$ leading to different observable functions and hence could be interpreted as different tasks to be meta-learned.

Specifically, when implementing PIED, we first use REPTILE to learn a shared PINN initialization $\theta_{\text{SI}}$ for the set of reference $\beta$, if we do not already have a pre-trained model for the system of interest[8]. Given this, we can then (1) efficiently fine-tune $\theta_{\text{SI}}$ across different reference $\beta$ with fewer training steps to produce the forward PINNs $\{\tilde{u}_{\beta_i}\}_{i=1}^{N}$, (2) re-use $\theta_{\text{SI}}$ for the approximation of inverse PINNs performance to optimize our ED criterion (elaborated in Sec. 4.2), and (3) re-use $\theta_{\text{SI}}$ for the final inverse PINN applied to actual experimental data to find the true $\hat{\beta}^*$ (Sec. 3.1). Once learnt, $\theta_{\text{SI}}$ can

---

[7]The observation input can either be freely chosen or constrained to certain configurations (see App. E).

[8]Note that no actual experimental data is needed for this pre-training phase, just like in the training of forward PINNs. Rather, training is done based on $\mathcal{L}_{\text{PDE}}(\theta, \beta)$ in (4), enforced at collocation points.

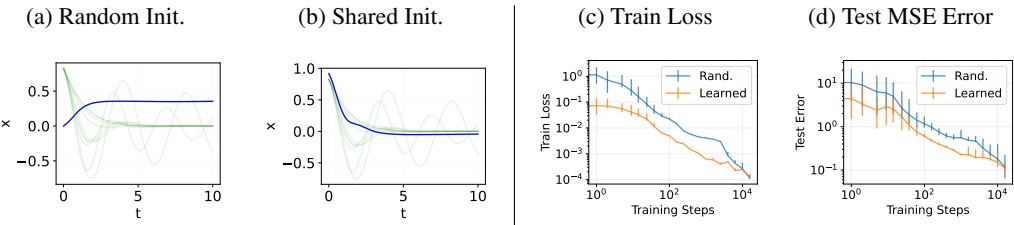

Figure 2: Results for meta-learning a shared NN initialization for PINNs trained on 1D damped oscillator case. The shared initialization (blue line) in Fig. 2b exhibits similar structure to PDE solutions $u_\beta$ for different values of $\beta$ (faint green lines), unlike the random initialization (blue line) in Fig. 2a. In Figs. 2c and 2d, we show that this translates to better average train and test loss performance of PINNs with shared initialization compared to random initialization w.r.t. different $\beta$.

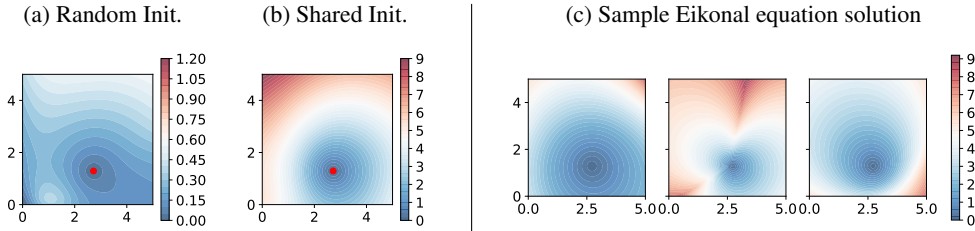

Figure 3: Results for learning a NN initialization for PINNs trained on Eikonal equation case. Fig. 3a represents a randomly initialized PINN, while Fig. 3b represents the shared initialzation for the PINN. Fig. 3c shows sample PDE solutions $u_\beta$ for different random PDE parameters $\beta$.

also be re-used many times on different IP instances with the same PDE setting, further reducing computational costs in practical applications.

To demonstrate the benefits of learning a shared NN initialization for PINNs, we visualize various PDE solutions (green lines) for the 1D damped oscillator setting, along with the output of the PINN with random initialization and shared initialization (blue lines), in Fig. 2a and Fig. 2b respectively. Note how the PINN with the meta-learned, shared initialization share qualitatively similar structure to the PDE solutions $u_\beta$ for different values of $\beta$, e.g., replicating the damping effect observed in the various solutions. In contrast, the random initialization in Fig. 2a is dissimilar to any of the PDE solutions. This advantage can also be observed quantitatively in Figs. 2c and 2d, where we plot the train and test loss for forward PINNs when initialized with $\theta_{\text{SI}}$ versus when initialized randomly. We can see that PINNs initialized with the shared $\theta_{\text{SI}}$ initialization have faster training convergence with lower train and test loss compared to those that are random initialized.

We also demonstrate this advantage for the 2D Eikonal equation setting, where the PDE parameters are functions of the input space $X$. In Fig. 3, we visualize the shared parameters learned for the Eikonal equation, which again shows that the qualitative structures and even the scaling of the PDE solutions can be meta-learned beforehand in order to speed up the PINN training process. We provide further empirical results in App. I.1, which demonstrates that this also applies for inverse PINNs and forward PINNs trained for other PDEs.

## 4.2 APPROXIMATE CRITERIA FOR PERFORMANCE OF THE INVERSE SOLVER

We now present two methods to effectively optimize for the observation inputs by approximating the inverse solver performance and criterion in (8).

**Few-step Inverse Solver Training (FIST) Criterion.** Our first method stems from the insight that we do not need to fully solve for $\hat{\beta}_i$ in PIED's inverse solvers to find the observation input $X$ that maximizes (8). Drawing inspiration from Lau et al. (2024a) with results showing that informative training points that lead to faster training convergence also result in lower PINN generalization error bounds and better empirical performance, we propose to only partially train the inverse PINN for

a few training steps to get an intermediate estimate of $\hat{\beta}_i$, and use this to perform gradient-based optimization for $X$ with (8) by back-propagating through the PINN inverse solver. To get more informative gradient signals, this is done with inverse PINNs that are closer to convergence rather than at early stages of training when most choices of $X$ would result in good performance gains.

Hence, our method FIST first initializes the inverse PINNs for each thread $i$ with perturbed NN parameters from the corresponding converged forward PINN for $\beta_i$, before partially training them for $r$ training steps to obtain the estimate $\hat{\beta}_i$, computing the criterion in (8), and performing gradient descent optimization for $X$. We present the pseudocode for FIST in Alg. 1 of the Appendix.

**Model Training Estimate (MoTE) Criterion.** Our second method involves directly approximating the inverse solver output $\hat{\beta}_i$ at convergence while minimizing training. We do so by performing kernel regression with the empirical Neural Tangent Kernel (eNTK) of the PINN (Jacot et al., 2018; Wang et al., 2022; Lau et al., 2024a), given a set of observation input $X$. Specifically, under assumptions that the PINN is in the linearized regime and trained via gradient descent with (4), the predicted PDE parameter $\hat{\beta}_i$ at convergence can be estimated by (IC/BC terms omitted for notational simplicity)

$$
\hat{\beta}_i(X, \tilde{Y}) \approx \beta^{(0)} - \begin{bmatrix} 0 \\ J_{p,\beta}^\top \end{bmatrix} \begin{bmatrix} J_{\text{obs},\theta} J_{\text{obs},\theta}^\top & J_{\text{obs},\theta} J_{p,\theta}^\top \\ J_{p,\theta} J_{\text{obs},\theta}^\top & J_{p,\theta} J_{p,\theta}^\top + J_{p,\beta} J_{p,\beta}^\top \end{bmatrix}^{-1} \begin{bmatrix} \hat{u}_{\theta^{(0)}}(X) - \tilde{Y} \\ \mathcal{D}[\hat{u}_{\theta^{(0)}}, \beta^{(0)}](X_p) - f(X_p) \end{bmatrix}
$$
(9)

where $(\theta^{(0)}, \hat{\beta}^{(0)})$ are the initialization parameters, $J_{\text{obs},\theta} = \nabla_\theta \hat{u}_{\theta^{(0)}}(X)$, $J_{p,\theta} = \nabla_\theta \mathcal{D}[\hat{u}_{\theta^{(0)}}, \beta](X_p)$, and $J_{p,\beta} = \nabla_\beta \mathcal{D}[\hat{u}_{\theta^{(0)}}, \beta^{(0)}](X_p)$ (note that $\nabla_\beta \hat{u}_{\theta^{(0)}}(X) = 0$). This estimate for $\hat{\beta}$ in (9) is adapted from Lee et al. (2018), and is verified in App. F.2.1, which includes further details on the assumptions and derivation. Note that while the use of NTK in PINNs is not new (Wang et al., 2022; Lau et al., 2024a), previous works have not utilized NTK to directly quantify the performance of PINNs in solving inverse problems as we have done.

In practice, as noted by Lau et al. (2024a), finite-width PINNs have eNTKs that evolve over training before they can better reconstruct the true PDE solution. Hence, for the MoTE criterion, we either first do $r$ steps of training on the $\theta_{\text{SI}}$ initialized inverse PINN before computing the eNTK and performing kernel regression, or use a perturbed version of the forward PINN parameter to perform kernel regression. We present the pseudocode for MoTE in Alg. 2 in the Appendix.

In App. F, we provide further discussion about the proposed criteria for approximating the inverse solver performances. In App. G, we summarize the full implementation process for PIED, incorporating both approximation methods and the full ED loop from Sec. 3.

## 5 RESULTS

In this section, we present empirical results to demonstrate the performance of PIED on ED problems for a range of scenarios based on different PDE systems, both from noisy simulations from real physical experiments (details of the specific PDEs are in App. D). For scenarios using simulation data, we injected noise and lowered data fidelity for the evaluation dataset to represent noisy limited sensor capabilities. We compare PIED against various benchmarks, such as approximations of the expected mutual information (Belghazi et al., 2018; Foster et al., 2019) and criterion from optimal sensor placement literature (Krause et al., 2008), in addition to random and grid-based methods, with details in App. H.4. In each scenario, the ED methods compute their optimal observation input, which are then evaluated on multiple IP instances (i.e., different ground truth PDE parameters $\beta$). We then report the mean error across the different PDE parameters in accordance to the loss term in (3). Additional details on the experimental setup are listed in App. H. We present a subset of experimental results in the main paper, and defer the remaining results to App. I. The code for the project can be found at `https://github.com/apivich-h/pied`.

**Finite-dimensional PDE parameters.** We first present two ED problems on IPs where the PDE parameters corresponds to multiple scalar terms representing certain physical properties of the system. The first scenario is the 1D time-dependent wave equation with inhomogeneous wave speeds, where we incorporated realistic elements such as restrictions in sensor placements. The second is the 2D Navier-Stokes equation, which is a challenging setting relevant to many important scientific and

| Dataset | Finite-dimensional | | Function-valued | Real dataset | |
|---|---|---|---|---|---|
| | Wave ($\times 10^{-1}$) | Navier-Stokes ($\times 10^{-2}$) | Eikonal ($\times 10^1$) | Groundwater ($\times 10^1$) | Cell Growth ($\times 10^0$) |
| Random | 5.23 (1.26) | 6.19 (3.53) | 1.82 (0.09) | 3.44 (1.77) | 3.63 (0.26) |
| Grid | 8.90 (0.73) | 4.51 (0.60) | 1.56 (0.22) | 2.27 (2.26) | 3.19 (0.23) |
| MI | 4.46 (0.92) | 6.08 (2.10) | 2.02 (0.01) | 2.10 (0.32) | 3.14 (0.59) |
| VBOED | 4.63 (1.64) | 4.33 (0.98) | 1.82 (0.50) | 2.29 (1.09) | 2.82 (1.77) |
| FIST (ours) | **3.87 (0.76)** | 2.10 (1.45) | **0.74 (0.02)** | **1.93 (0.08)** | **2.62 (0.11)** |
| MoTE (ours) | **3.81 (2.34)** | **1.18 (0.11)** | **0.76 (0.02)** | **2.00 (0.60)** | 2.83 (0.04) |

Table 1: Results of the ED methods for the various experimental scenarios. Each result reports the median of the expected loss (i.e., in (3)) across trials, and the figure in bracket represents the semi-interquartile range. The results of the best performing ED methods in each dataset are in bold.

industrial problems. The results are presented in Table 1. Compared to benchmarks, both FIST and MoTE perform better in the scenarios, producing optimal $X$ choices that have lower expected loss when evaluated across datasets of multiple $\beta$ values. The performance gap between our methods and benchmarks is more significant for the 2D Navier-Stokes scenario, which is more challenging, making the optimization of $X$ more important in obtaining good estimates of the inverse problem.

**PDE parameters that are functions of input space.** We further demonstrate that PIED can be applied to complex scenarios where the PDE parameter of interest $\beta$ is a function defined over the input space $\mathcal{X}$. To estimate $\beta$ in these cases, we parameterize it using a small NN, and learn it together with the PINNs in our framework. The scenario we considered is the 2D Eikonal equation, which is used in a wide range of applications such as seismology, robotics or image processing. In this problem, the sensors can be placed freely in $\mathcal{X}$, yielding a high-dimensional design parameter. The results in Table 1 demonstrate that even in this complex scenario, our PIED methods are still able to outperform the benchmarks and recover the correct function-valued PDE parameter. In addition, our methods also produce consistently good results, as can be seen from the small semi-interquartile range (SIQR) of our expected loss.

We can further analyze the performance of PIED by visualizing the observation inputs $X$ chosen by FIST and the loss of the PDE parameter (Fig. 4). Note that FIST automatically adjusts the choice of $X$ based on underlying structure of the problem, allowing it to choose $X$ to more evenly minimize the error of $\beta$. This provides it with advantages over more heuristics-based approaches like the space-filling Grid method.

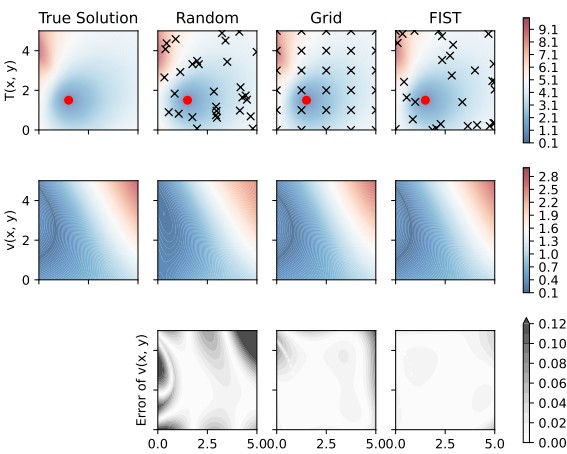

Figure 4: Example of ED process involving the Eikonal equation using different methods of observation selection. Top row: the true observation $T(x, y)$ and its approximation via PINNs. Middle row: the true unknown function $v(x, y)$ and the recovered estimations based on observation inputs. Bottom row: the error of the reconstructed $v(x, y)$.

**Inverse problem on real-life dataset.** Simulation datasets may not be able to fully represent the challenges of realistic scenarios, due to the complexity of the real-life noisy experimental data. Hence, to further validate the advantages of PIED and analyze its performance on more realistic scenarios, we applied it to observation selection problems on *real data* that are collected from physical experiments in different domains of natural science. Specifically, we consider the ED problem applied to the groundwater flow dataset collated by Shadab et al. (2023), and the scratch assay cell population growth data collected by Jin et al. (2016), where the design parameters indicate the location to make the observations at constrained, fixed time intervals. As can be seen in Table 1, our PIED methods outperform benchmarks in both scenarios, despite the challenges posed by the datasets. For example, we can see in Fig. 5a that the groundwater flow dataset no longer fits typical i.i.d. noise assumptions that existing ED method work under, but FIST still performs well (selected observations

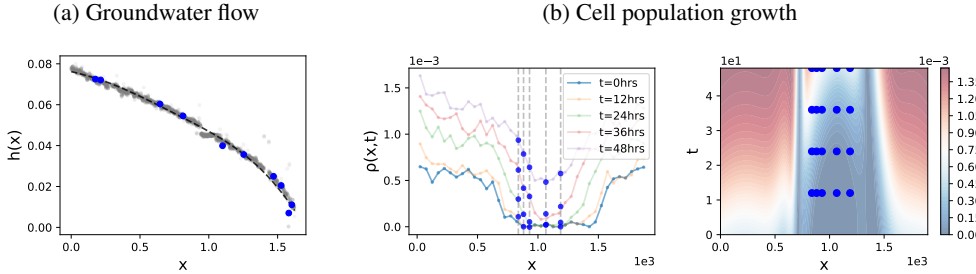

Figure 5: Visualization of real-life experimental data used in our tests, along with demonstration of observations selected by FIST. Fig. 5a: example of groundwater flow data from Shadab et al. (2023). Gray points represent the collected data, while the blue points are the observations chosen by FIST. The black line represents the prediction from the corresponding inverse PINN. Fig. 5b: example of cell population growth data from Jin et al. (2016). The left figure shows the cell population data which are collected at 12 hour intervals, while the right figure shows the population prediction from the inverse PINN. In both figures, the blue points represent the observations chosen by FIST.

in blue, and IP prediction plotted as a dotted line). In Fig. 5b, we can see that the observations chosen by FIST are spatially close to each other in order to better interpolate the effects from spatially-related PDE parameters, resulting in its better performance compared to benchmarks.

**Advantages of PINNs over numerical solvers in ED.** In the results above, we implemented the benchmark ED methods applied to PINNs as forward simulators. This allowed the methods to benefit from the advantages of PINNs. In fact, existing benchmarks using classical numerical solvers would produce significantly worse results than those reflected in Table 1. To see this, we consider the 2D Eikonal scenario, and use the Eikonal equation solver as implemented by White et al. (2020) which uses the fast marching method (Sethian, 1996) as the numerical simulator. Due to the lack of gradient information, optimization is done using Bayesian optimization. Fig. 6a shows that using PINNs allow the correct function to be recovered much more accurately regardless of the ED method used. Furthermore, we see in Fig. 6b that using PINNs allow the overall ED and IP to be done more efficiently as well.

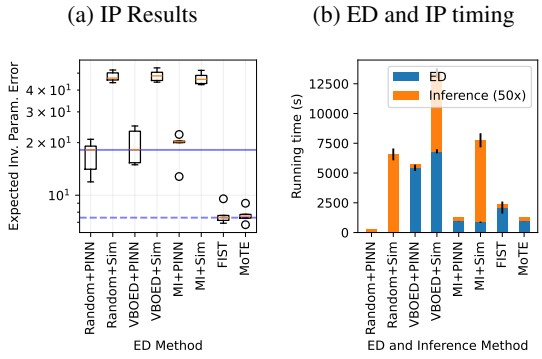

Figure 6: Results for various benchmarks using numerical simulators and PINNs. Fig. 6a: error of recovered PDE parameters. The thick blue line represents the performance of the Random method, while the dashed line represents the best performance. Fig. 6b: time required to run the ED algorithm and to run 50 rounds of inference to solve the inverse problem.

## 6 CONCLUSION

We have introduced PIED, the first ED framework that utilizes PINNs as both forward simulators and inverse solvers in a fully differentiable architecture to perform continuous optimization of design parameters for IPs. PIED selects optimal design parameters for one-shot deployment, and allows exploitation of parallel computation unlike existing methods. We have also designed effective criteria for the framework which are end-to-end differentiable and hence can be optimized through gradient-based methods. Future work could include applying PIED to other differentiable physics-informed architectures, such as operator learning methods.

## ACKNOWLEDGMENTS

This research/project is supported by the National Research Foundation, Singapore under its AI Singapore Programme (AISG Award No: AISG2-PhD/2023-01-039J) and is part of the programme DesCartes which is supported by the National Research Foundation, Prime Minister's Office, Singapore under its Campus for Research Excellence and Technological Enterprise (CREATE) programme. The computational work for this article was partially performed on resources of the National Supercomputing Centre, Singapore (https://www.nscc.sg).

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

## A    REPRODUCIBILITY STATEMENT

The codes for our implementation, scripts for running experiments and the required dataset are attached in the supplementary materials of the paper submission.

## B    NOTATIONS

Table 2: List of notations used throughout the paper

| Symbol | Meaning | Example |
|---|---|---|
| $\mathcal{D}$ | PDE operator | (1) |
| $\mathcal{B}$ | Boundary condition operator | (1) |
| $\mathcal{X}$ | Input domain | (1) |
| $\partial \mathcal{X}$ | Boundary of input domain | (1) |
| $\mathcal{S}$ | Set of feasible observation inputs | |
| $\beta$ | PDE parameter | (1) |
| $u_\beta$ | Solution for (1) with PDE parameter $\beta$ | (2) |
| $\hat{\beta}$ | Estimate of PDE parameter from inverse solver | (2) |
| $\theta$ | NN parameter | (4) |
| $\hat{u}_\theta$ | NN with parameters $\theta$ | (4) |
| $\mathcal{L}$ | PINN training loss | (4) |
| $\mathcal{L}_{\text{obs}}$ | Observation loss for PINN | (4) |
| $\mathcal{L}_{\text{PDE}}$ | Collocation points loss for PINN | (4) |
| $\mathbf{F}$ | Forward simulator | (5) |
| $\tilde{u}_{\beta_i}$ | PINN, with NN parameter $\theta_i$, which estimates $u_{\beta_i}$ | (5) |
| $\mathbf{O}_X$ | Observation selector with input $X$ | (6) |
| $\tilde{Y}$ | Mock observation output from $\tilde{u}_{\beta_i}(X)$ | (6) |
| $\mathbf{I}$ | Inverse solver | (7) |
| $\alpha$ | ED criterion without inverse ensemble approximation | (8) |
| $\alpha_i$ | ED criterion computed based on outputs of thread $i$ of framework | (8) |
| $\gamma$ | Parameterization for observation input | (23) |
| $X_\gamma$ | Observation input corresponding to parameter $\gamma$ | (23) |
| $\mathcal{S}_\gamma$ | Set of valid observation input parameterization | (23) |
| $\nabla_x$ | Derivative or Jacobian w.r.t. $x$ | |
| $\nabla_x^2$ | Hessian w.r.t. $x$ | |
| $[a, b]$ | Closed interval between $a$ and $b$ | |

## C    RELATED WORKS

**Inverse problems.**    Inverse problem (IP) (Vogel, 2002; Ghattas & Willcox, 2021) is an commonly studied class of problem in many science and engineering disciplines such as classical mechanics (Tanaka & Bui, 1993; Gazzola et al., 2018), quantum mechanics (Chadan et al., 1989) or geophysics (Smith et al., 2021; Waheed et al., 2021). Many methods of solving IPs have been proposed, often involving minimizing the objective as stated in (2), possibly with addition of some regularization terms. One such method is by using the Newton-conjugate gradient method (Biegler et al., 2003; Ghattas & Willcox, 2021) to optimize the objective as stated in (2). However, the method relies on finding the optimal $\beta$ through gradient update steps, which requires computing the gradient and Hessian of the objective function with respect to $\beta$. The computation of the gradient and Hessian is often done by reformulating the IP (which can be viewed as a constrained optimization problem) to instead be based on the Lagrangian, then using adjoint methods to compute the corresponding gradient or Hessian (Ghattas & Willcox, 2021). This typically results in gradient and Hessian computations requiring only some finite rounds of forward simulations instead. The computed Hessian can often also be used in Laplace's approximation in order to obtain a posterior distribution $p(\beta|X, Y)$ for the inverse parameter (Long et al., 2013; Beck et al., 2018; Ghattas & Willcox, 2021). However, this method is still restrictive since it may involve careful analysis of the PDE that is involved in the IP in order to form the correct Lagrangian and compute the adjoint. Furthermore, one gradient computation would require one forward simulation of the system, which is prohibitive if the forward simulation step itself is expensive.

**Physics-informed neural networks.** In recent years, physics-informed neural networks (PINNs) have been proposed as another method used to both perform forward simulations of PDE-based problems (Raissi et al., 2019) and for solving IPs (Raissi et al., 2019). PINNs solve PDEs by parameterizing the PDE solution using a neural network (NN), then finding the NN parameter such that the resulting NN obeys the specified PDE and the IC/BCs. This is done so via collocation points, which are pseudo-training points for enforcing the PDE and IC/BC soft constraints. PINNs are difficult to train in many PDE instances, such as when the solution is known to have higher frequencies (Wang et al., 2022). As a result, many works have been proposed in improving the trianing of PINNs by rescaling the loss functions of PDEs (Wang et al., 2022), or through more careful selection of collocation points (Wu et al., 2023; Lau et al., 2024a).

**Experimental design.** Typically, in solving IPs, the observational data will not be available right away, but instead has to be measured from some physical system. Due to the costs in making measurements of data, it is important that the observations made are carefully chosen to maximize the amount of information that can be obtained from the observations. Experimental design (ED) is a problem which attempts to find out what the best data to observe in order to gather the most information about the unknown quantity of interest (Rainforth et al., 2023). ED is closely related to data selection, active learning (Nguyen et al., 2021; Hemachandra et al., 2023; Lau et al., 2024b; Xu et al., 2023) and Bayesian optimization (Dai et al., 2023a;b; Chen et al., 2025), where the aim is to select input data so that we are able to recover the unknown function or the optimum of the unknown function.

A certain variant often considered in ED is Bayesian experimental design (BED). In the BED framework, we assume a prior $p(\beta)$ on the inverse parameter to compute. For a given design parameter $d$, the system observes some output $y$.

$$p(\beta|d, y) = \frac{p(y|\beta, d)\, p(\beta)}{\mathbb{E}_{\beta \sim p(\beta)}\big[p(y|\beta, d)\big]}. \tag{10}$$

Given the inference we can compute the expected information gain (EIG), which is sometimes also known as the Bayesian D-optimal criterion. EIG criterion is defined as the expected Kullback-Leibler divergence between the prior $p(\beta)$ and the posterior $p(\beta|d, y)$, averaged over the possible observations $y$. More formally, this can be written as

$$\text{EIG}(d) = \mathbb{E}_{y \sim p(y|d)}\big[D_{\text{KL}}\big(p(\beta|d, y)\|p(\beta)\big)\big] = \mathbb{H}[p(\beta)] - \mathbb{E}_{y' \sim p(y|d)}\big[\mathbb{H}[p(\beta|d, y = y')]\big] \tag{11}$$

where the posterior is defined in (10). A naive approximation technique is to perform a nested Monte Carlo (NMC) approximation (Myung et al., 2013).

$$\text{EIG}(d) \approx \frac{1}{N} \sum_{i=1}^{N} \log \frac{p(y_i|\beta_{i,0}, d)}{\frac{1}{M} \sum_{j=1}^{M} p(y_i|\beta_{i,j}, d)} \tag{12}$$

where $\beta_{i,0}, \beta_{i,1}, \ldots, \beta_{i,j} \sim p(\beta)$ and $y_i \sim p(y|\beta_{i,0}, d)$. The estimate approaches the true EIG as $N, M \to \infty$. In practice, however, using the NMC estimator results in a biased estimtor for finite $N$ and $M$, and results in slow convergence with $N$ and $M$. To improve on the NMC estimator, various schemes have been proposed mainly to remove the need to perform two nested MC rounds, including variational methods (Foster et al., 2019) and Laplace approximation methods (Long et al., 2015; Beck et al., 2018).

## D EXAMPLES OF PDEs CONSIDERED IN THIS PAPER

In this section, we provide an extensive list of PDEs which are considered in our experimental setup, and the ED setup and dataset used in our experiments. We divide them into PDEs where our experiments are based on simulation data (generated either from some closed-form solution or some numerical simulators), and PDEs which are based on real data (collected from physical experiments). The latter provides an interesting use case for PIED since it is able to demonstrate its performance on realistic scenarios with real noisy data.

### D.1 PDEs with Simulation Data

**Damped oscillator.** The damped oscillator is one of the introductory second-order ordinary differential equation (ODE) in classical mechanics (Taylor, 2005). We consider the example due to the existence of a closed-form solution and its nice interpretation under our ED framework.

Imagine a mass-spring system which is laid horizontally. The spring has spring constant $k$ and experiences a resistive force which is proportional to its current speed, where the constant of proportionality is $\mu$. We let the attached mass have a mass of $M$. We also assume the case where there are no external driving forces on the system. By applying the relevant forces into Newton's law of motion, the displacement of the mass $x(t)$ can be expressed by the differential equation

$$M\frac{d^2x}{dt^2} + \mu\frac{dx}{dt} + kx = 0. \tag{13}$$

Given the IC of $x(0) = x_0$ and $\frac{dx}{dt}(0) = v_0$, we can write the solution as (Taylor, 2005)

$$x(t) = \begin{cases} Ae^{-\gamma t}\cos(\sqrt{\omega_0^2 - \gamma^2}t + \phi) & \text{if } \gamma < \omega_0, \\ (Bt + C)e^{-\omega_0 t} & \text{if } \gamma = \omega_0, \\ De^{-(\gamma + \sqrt{\gamma^2 - \omega_0^2})t} + Fe^{-(\gamma - \sqrt{\gamma^2 - \omega_0^2})t} & \text{if } \gamma > \omega_0, \end{cases} \tag{14}$$

where $\gamma = \mu/2M$, $\omega_0 = \sqrt{k/M}$, and $A, \phi, B, C, D, F$ are constants which depends on $x_0$ and $v_0$.

In our experiments, we assume we know the system follows the PDE as in (13), with some known value of $x_0 \in [0, 1]$ and $v_0 \in [-1, 1]$. We set $M = 1$, and would like to compute the values for $\mu \in [0, 4]$ and $k \in [0, 4]$. In our IP, we are allowed to make three noisy observations at three timesteps $t_1, t_2, t_3 \in [0, 20]$, which can be chosen arbitrarily. We note that while it is unrealistic for these measurements to be made arbitrarily, we do so in order to be able to construct a simple toy example which can be experimented with. The resulting true observation were computed using the closed form solution in (14), and has added noise with variance $10^{-3}$.

**Wave equation.** For simplicity, we consider the 1D wave equation, which is given by

$$\frac{\partial^2 u}{\partial t^2} = v^2 \frac{\partial^2 u}{\partial x^2} \tag{15}$$

where $v$ represents the speed of wave propagation, which may be a scalar or a function of $x$. In the inevrse problem setup, one may be required to recover the wave velocity $v$ given measurements of $u(x, t)$.

In our experiments, we assume we have a system which follows the wave equation given by (15) over the domain $x \in [0, 6]$ and $t \in [0, 6]$. We fix the IC $u(x, 0)$, and assume the wave velocity in the form

$$v(x) = \begin{cases} 0 & \text{if } x = 0, \\ v_1 & \text{if } 0 < x < 4, \\ v_2 & \text{if } 4 \le x < 6, \\ 0 & \text{if } x = 6. \end{cases} \tag{16}$$

In this case, $v_1, v_2 \in [0.5, 2]$ are the PDE parameters to be recovered in the IP. In the ED problem, we restrict the points to be placed only at regular time intervals, i.e., following (25) where we restrict $\gamma_1, \gamma_2, \gamma_3 \in [0, 6]$ and let $t \in \{0, 0.2, 0.4, \ldots, 6\}$. The true observations are numerically generated using code from Binder (2021), with outputs interpolated on continuous domain and truncated to the nearest 6 decimal places to simulate cases where measurements can only be made up to a finite precision. Examples of these possible solutions can be found in Fig. 7.

**Navier-Stokes equation.** Navier-Stokes equation is a well-studied PDE which describes the dynamics of a fluid. In our experiment, we consider the stream function of an incompressible 2D fluid,

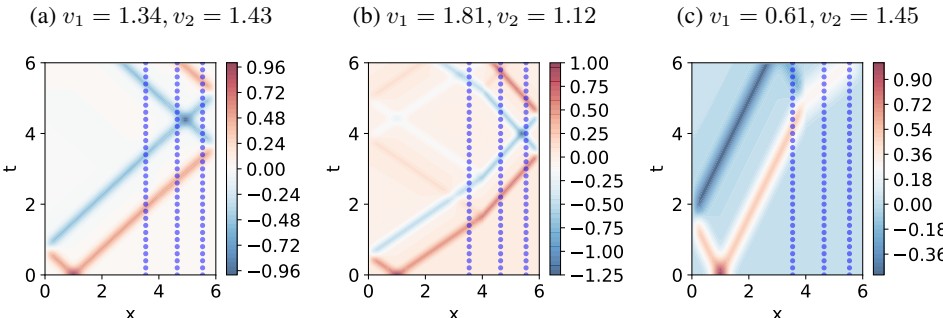

Figure 7: Examples of solutions for wave equation (15) with wavespeed in the form in (16). The blue points are example of possible set of observations which are made at fixed timesteps.

which can be written as

$$\frac{\partial u}{\partial t} + \rho\left(u\frac{\partial u}{\partial x} + v\frac{\partial u}{\partial y}\right) = -\frac{\partial p}{\partial x} + \mu\left(\frac{\partial^2 u}{\partial x^2} + \frac{\partial^2 u}{\partial y^2}\right), \tag{17}$$

$$\frac{\partial v}{\partial t} + \rho\left(u\frac{\partial v}{\partial x} + v\frac{\partial v}{\partial y}\right) = -\frac{\partial p}{\partial y} + \mu\left(\frac{\partial^2 v}{\partial x^2} + \frac{\partial^2 v}{\partial y^2}\right), \tag{18}$$

$$\frac{\partial u}{\partial x} + \frac{\partial v}{\partial y} = 0. \tag{19}$$

In our IP, we consider the steady state flow inside a pipe, where we let $\partial_t u = \partial_t v = 0$. Then, the velocities $u(x, y), v(x, y)$ and the pressure $p(x, y)$ are only dependent on the 2D spatial coordinates. We assume the viscosity $\mu$ is given and the goal is to recover the density $\rho$. In the ED problem, we are allowed to freely choose the spatial location to make measurements of $u, v, p$. The ground truth data is simulated using ANSYS Fluent.

**Eikonal equation.** Consider the Eikonal problem setup, which is often use to reconstruct material composition of some region based on how waves propagated through the medium reacts. Its equation relates the wave speed $v(x)$ at a point and the wave propagation time $T(x)$ at a point with PDE given by (Smith et al., 2021)

$$T(x) = \left(\nabla v(x)\right)^{-1} \quad \text{with} \quad T(x_0) = 0 \tag{20}$$

where $x_0$ is where the wave propagates from. The goal of the IP is to recover the true function $v$. However, this involves conducting seismic activities at different set values of $x_0$, and obtaining the corresponding reading for $T(x)$ at specified values of $x$.

In our experiments, we assume that we have a system which follows the equation specified in (20), and the goal is to recover the values of $v(x)$ for the entire domain. We fix $x_0$, then for each IP instance, we draw a random ground truth $v(x)$ using a NN with a random initialization. For the ED problem, the aim is to find 30 random observations from the 2D input domain $[0, 5] \times [0, 5]$ to observe values of $T(x)$. The observation inputs are only required to be within the input domain. The true observations are generated using PYKONAL package (White et al., 2020), with outputs interpolated on continuous domain and truncated to the nearest 3 decimal places to simulate cases where measurements can only be made up to a finite precision

### D.2 PDEs WITH REAL DATA

We now describe some of the PDEs we have conducted experiments with where real-life data are available for. Note that real-life experimental data is often scarce, and often the true PDE parameters are not readily available. To obtain some ground-truth values for the inverse problem, we attempt to use the reported values from the corresponding data source when we can. In the case where this is not possible, we resort to numerically computing the PDE parameters using the whole training set.

**Groundwater flow.** In our scenario, we consider the steady-state Dupuit-Boussinesq equation (Boussinesq, 1904), which is given by

$$\frac{d}{dx}\left(Kh\frac{dh}{dx}\right) = 0 \tag{21}$$

where $K$ is the hydraulic conductivity.

In our experiments, we use the data provided in (Shadab et al., 2023), which reports the flow profile of some liquid at various flow rates across a cell filled with 2mm beads. In this dataset, we treat the hydraulic conductivity $K$ as an unknown quantity we would like to recover in our inverse problem. In the ED problem, the algorithms have to choose values of $x$ to make the observations $h(x)$.

**Cell growth.** Scratch assay experiments can often be modelled via a reaction-diffusion PDE (Jin et al., 2016), which can be written as

$$\frac{\partial \rho}{\partial t} = c_1 \frac{\partial^2 \rho}{\partial x^2} + F[\rho] \tag{22}$$

where $F[\rho]$ is some function of $\rho$ and $c_1$ is an unknown constant.

In our experiments, we use the scratch assay data collected by Jin et al. (2016), whose dynamics have also been studied in other subsequent papers Lagergren et al. (2020); Chen et al. (2021). Based on Chen et al. (2021), we will let $F[\rho] = c_2\rho + c_3\rho^2$ in our case, where $c_2$ and $c_3$ are unknown. The data is collected at five timesteps every 12 hours for a total of 48 hours. We use the values obtained by Chen et al. (2021) as the ground truth values for $c_1, c_2, c_3$. In our ED problem, the IP with the initial population values $\rho(x, t=0)$ are given, and the algorithms choose the values of $x$ to query the cell population at, where the corresponding values will be provided at each of the other four timesteps.

## E   PARAMETERIZATION OF INPUT POINTS

Due to operational constraints, the observation input $X$ often cannot be set arbitrarily, but is instead restricted to some feasible set $\mathcal{S} \subset \mathcal{X}^M$. For example, the observations may be only be possible at specific time intervals, or must be placed in certain spatial configurations like a regular grid. Hence, unlike existing ED works, we allow for both freely-chosen observation inputs and *constrained configurations* which can be parameterized by some design parameter $\gamma \in \mathcal{S}_\gamma \subset \mathbb{R}^d$ where $d \leq Md_{\text{in}}$. Specifically, we consider $\mathcal{S}$ of the form

$$\mathcal{S} = \left\{X_\gamma : \gamma \in \mathcal{S}_\gamma\right\} \quad \text{where} \quad X_\gamma = \left\{x_{\gamma,j}\right\}_{j=1}^M. \tag{23}$$

In this form, finding the $X$ which optimizes some criterion is then the same as performing optimization on $\gamma$ instead, which can be seen as a continuous optimization problem.

To better illustrate the input points parameterization method, we provide some examples of possible methods to constrain the input points and how they be expressed in the appropriate forms. Fig. 8 graphically demonstrate what some of these observation input constraints may look like.

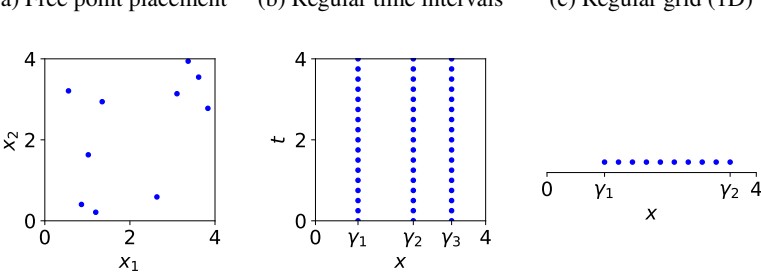

Figure 8: Examples of observation input placements.

**Points placed freely in input space.** In this case, the points can be placed anywhere in $\mathcal{X}$ without restriction. To parameterize this, we define $d = M d_{\text{in}}$, and define

$$X_\gamma = \Big\{ \big( \gamma_1 \ \gamma_2 \ \cdots \ \gamma_{d_{\text{in}}} \big), \ \big( \gamma_{d_{\text{in}}+1} \ \gamma_{d_{\text{in}}+2} \ \cdots \ \gamma_{2d_{\text{in}}} \big), \ \ldots, \ \big( \gamma_{M d_{\text{in}}-d_{\text{in}}} \ \gamma_{M d_{\text{in}}-d_{\text{in}}+1} \ \cdots \ \gamma_{M d_{\text{in}}} \big) \Big\}$$

(24)

**Points placed at regular time intervals.** In this case, we assume the points are placed at chosen spatial locations, and makes measurements at fixed time intervals $t_1, t_2, \ldots, t_f$. This is realistic in the case where the PDE solution evolves over time, and so it makes sense to fix the location of the sensor but allow it to make readings throughout the evolution of the system over time. In this case, $\gamma$ only needs to encode the spatial location where the sensors should be placed. Specifically, if there is one spatial dimension, then the parameterization for the sensors can be chosen as

$$X_\gamma = \Big\{ (x, t) : x \in \big\{ \gamma_1, \gamma_2, \ldots, \gamma_d \big\} \text{ and } t \in \{ t_1, t_2, \ldots, t_f \} \Big\}.$$

(25)

**Points placed in a regular grid.** In this case, the points are placed in a regular grid at regular intervals. This provides one way to add extra constraints for sensor configurations to reduce the dimension of the problem. For demonstration, in 1D problems, if we want to allow placement of $s$ sensors in total, we can let $d = 2$ and parameterize the sensor placements as

$$X_\gamma = \Big\{ \gamma_1, \ \gamma_1 + \frac{\gamma_2 - \gamma_1}{s - 2}, \ \gamma_1 + 2 \cdot \frac{\gamma_2 - \gamma_1}{s - 2}, \ \ldots, \ \gamma_2 \Big\}.$$

(26)

# F  FURTHER DETAILS ABOUT THE EXPERIMENTAL DESIGN CRITERION

We describe the criteria used further. Note that we adjust the notations to incorporate the parameterization of input as discussed in App. E.

## F.1  FEW-STEP INVERSE SOLVER TRAINING CRITERION

### F.1.1  PSEUDOCODE FOR CRITERION COMPUTATION

---

**Algorithm 1** Criterion estimation by Few-step Inverse Solver Training

---

1: **function** $\hat{\alpha}_{\text{FIST},i}(X_\gamma)$
2:     // Perturbation of NN and estimated PDE parameters
3:     $\bar{\theta} \leftarrow \theta_i + \varepsilon_\theta$ where $\varepsilon_\theta \sim \mathcal{N}(0, \sigma^2)$
4:     $\bar{\beta} \leftarrow \beta_i + \varepsilon_\beta$ where $\varepsilon_\beta \sim \mathcal{N}(0, \sigma^2)$
5:     $\tilde{Y}_\gamma \leftarrow \tilde{u}_{\bar{\beta}_i}(X_\gamma)$
6:     // Partial training stage
7:     Initialize $(\theta^{(0)}, \beta^{(0)})$
8:     **for** $j = 1, \ldots, r$ **do**
9:         // The training may be replaced with other gradient-based methods as well in practice
10:         $\theta^{(j)} \leftarrow \theta^{(j-1)} - \eta \nabla_\theta \mathcal{L}(\theta^{(j-1)}, \hat{\beta}^{(j-1)}; X_\gamma, \tilde{Y}_\gamma)$
11:         $\hat{\beta}^{(j)} \leftarrow \hat{\beta}^{(j-1)} - \eta \nabla_\beta \mathcal{L}(\theta^{(j-1)}, \hat{\beta}^{(j-1)}; X_\gamma, \tilde{Y}_\gamma)$
12:     **return** $-\|\hat{\beta}^{(r)} - \beta_i\|^2$

---

Furthermore, a large value of $r$ can also make the criterion not differentiable in practice due to the need to perform back-propagation over the gradient descent update steps. Fortunately, we find that in our experiments, using $r \leq 200$ is usually sufficient given the correct perturbation noise level is set.

## F.2  MODEL TRAINING ESTIMATE CRITERION

### F.2.1  ASSUMPTIONS AND PROOF OF (9)

In this section, we describe the approximation of training dynamics used in Model Training Estimate, which relies on approximation of NN training using NTKs (Jacot et al., 2018) and has been used

extensively in various active learning and data valuation methods (Wu et al., 2022; Hemachandra et al., 2023; Lau et al., 2024a).

We consider PINNs in the linearized regime to demonstrate the validity of the approximation given in (9). The results will be an extension from that given in (Lee et al., 2019), and is an assumption that has been used in past PINN works (Lau et al., 2024a). For convenience, we will drop the subscript and write the learnable PDE parameter as $\hat{\beta}$ and the NN parameters as $\theta$.

We first recall the assumptions for the linearized regime of NNs. Following past works on the NTK for NNs (Jacot et al., 2018; Lee et al., 2019) and PINNs (Lau et al., 2024a), we assume that

$$\hat{u}_\theta(x) \approx \hat{u}_{\theta^{(0)}}(x) + J_{\gamma,\theta}^{(0)}(\theta - \theta^{(0)}) \tag{27}$$

and

$$\mathcal{D}[\hat{u}_\theta, \hat{\beta}](x) \approx \mathcal{D}[\hat{u}_{\theta^{(0)}}, \hat{\beta}^{(0)}](x) + \begin{bmatrix} J_{p,\hat{\beta}}^{(0)} & J_{p,\theta}^{(0)} \end{bmatrix} \begin{bmatrix} \hat{\beta} - \hat{\beta}^{(0)} \\ \theta - \theta^{(0)} \end{bmatrix}. \tag{28}$$

We briefly discuss the consequences of this approximation. The linearized regime only holds when the learned parameters are similar to the initial parameters, which based on past works will hold when the NN is wide enough or when the NN is near convergence. This is unlikely to hold in the real training of PINNs, except in the case where the initialized parameters are already close to the true converged parameters anyway. Nonetheless, in our work, we do not use the assumptions to make actual predictions on the PDE parameters in the IP, however only use it to predict the direction of descent for $\gamma$, which would only use the local values anyway. Furthermore, we can also perform some pre-training in order to get closer to the converged parameters first as well to make the assumptions more valid.

Let $\hat{\beta}^{(t)}$ and $\theta^{(t)}$ be the learned PDE parameter and NN parameter respectively at step $t$ of the GD training. We will write $J_{\gamma,\theta}^{(t)} = \nabla_\theta \hat{u}_{\theta^{(t)}}(X_\gamma)$, $J_{p,\theta}^{(t)} = \nabla_\theta \mathcal{D}[\hat{u}_{\theta^{(t)}}, \hat{\beta}^{(t)}](X_p)$, and $J_{p,\hat{\beta}}^{(t)} = \nabla_\beta \mathcal{D}[\hat{u}_{\theta^{(t)}}, \hat{\beta}^{(t)}](X_p)$ (note that $\nabla_\beta \hat{u}_{\theta^{(t)}}(X_\gamma) = 0$). We can then write the GD training step as

$$\frac{\partial}{\partial t} \begin{bmatrix} \hat{\beta}^{(t)} \\ \theta^{(t)} \end{bmatrix} = -\eta \begin{bmatrix} \nabla_\beta \mathcal{L}(\theta^{(t)}, \hat{\beta}^{(t)}; X_\gamma, Y) \\ \nabla_\theta \mathcal{L}(\theta^{(t)}, \hat{\beta}^{(t)}; X_\gamma, Y) \end{bmatrix} \tag{29}$$

$$= -\eta \underbrace{\begin{bmatrix} 0 & J_{p,\beta}^{(t)} \\ J_{\gamma,\theta}^{(t)} & J_{p,\theta}^{(t)} \end{bmatrix}}_{\mathcal{J}_\gamma^{(t)}} \begin{bmatrix} \hat{u}_{\theta^{(t)}}(X_\gamma) - \tilde{Y}_\gamma \\ \mathcal{D}[\hat{u}_{\theta^{(t)}}, \hat{\beta}^{(t)}](X_p) - f(X_p) \end{bmatrix}. \tag{30}$$

Under the linearized regime, we can see that, $J_{\gamma,\theta}^{(t)} \approx J_{\gamma,\theta}^{(0)}$, $J_{p,\hat{\beta}}^{(t)} \approx J_{p,\hat{\beta}}^{(0)}$ and $J_{p,\theta}^{(t)} \approx J_{p,\theta}^{(0)}$. We can use these approximations to obtain

$$\frac{\partial}{\partial t} \begin{bmatrix} \hat{\beta}^{(t)} - \hat{\beta}^{(0)} \\ \theta^{(t)} - \theta^{(0)} \end{bmatrix} = \frac{\partial}{\partial t} \begin{bmatrix} \hat{\beta}^{(t)} \\ \theta^{(t)} \end{bmatrix} \tag{31}$$

$$\approx -\eta \mathcal{J}_\gamma^{(0)} \mathcal{J}_\gamma^{(0)\top} \begin{bmatrix} \hat{\beta}^{(t)} - \hat{\beta}^{(0)} \\ \theta^{(t)} - \theta^{(0)} \end{bmatrix} - \eta \mathcal{J}_\gamma^{(0)} \begin{bmatrix} \hat{u}_{\theta^{(0)}}(X_\gamma) - \tilde{Y}_\gamma \\ \mathcal{D}[\hat{u}_{\theta^{(0)}}, \hat{\beta}^{(0)}](X_p) - f(X_p) \end{bmatrix} \tag{32}$$

which can be solved to give

$$\begin{bmatrix} \hat{\beta}^{(t)} - \hat{\beta}^{(0)} \\ \theta^{(t)} - \theta^{(0)} \end{bmatrix} = -\mathcal{J}_\gamma^{(0)\top} (\mathcal{J}_\gamma^{(0)} \mathcal{J}_\gamma^{(0)\top})^{-1} \left(I - e^{-\eta \mathcal{J}_\gamma^{(0)} \mathcal{J}_\gamma^{(0)\top} t}\right) \begin{bmatrix} \hat{u}_{\theta^{(0)}}(X_\gamma) - \tilde{Y}_\gamma \\ \mathcal{D}[\hat{u}_{\theta^{(0)}}, \hat{\beta}^{(0)}](X_p) - f(X_p) \end{bmatrix}. \tag{33}$$

At convergence, i.e., when $t \to \infty$, we can reduce the results for $\hat{\beta}^{(\infty)} - \hat{\beta}^{(0)}$ as

$$\hat{\beta}^{(\infty)} - \hat{\beta}^{(0)} \approx \begin{bmatrix} 0 \\ J_{p,\beta}^{(0)\top} \end{bmatrix} (\mathcal{J}_\gamma^{(0)} \mathcal{J}_\gamma^{(0)\top})^{-1} \begin{bmatrix} \hat{u}_{\theta^{(0)}}(X_\gamma) - \tilde{Y}_\gamma \\ \mathcal{D}[\hat{u}_{\theta^{(0)}}, \hat{\beta}^{(0)}](X_p) - f(X_p) \end{bmatrix} \tag{34}$$

as claimed in (9).

### F.2.2 PSEUDOCODE FOR CRITERION COMPUTATION

---

**Algorithm 2** Criterion estimation by Model Training Estimate

---

1: **function** $\hat{\alpha}_{\mathrm{MoTE},i}(\gamma)$
2:     $\tilde{Y}_\gamma \leftarrow \tilde{u}_{\beta_i}(X_\gamma)$
3:     **if** reuse forward PINN parameters **then**
4:        $\theta^{(j)} \leftarrow$ perturbed version of forward PINN parameters
5:        $\hat{\beta}^{(j)} \leftarrow$ perturbed version of $\beta_i$
6:     **else**
7:        Initialize $(\theta^{(0)}, \beta^{(0)})$               ▷ Can set $\theta^{(0)}$ to $\theta_{\mathrm{SI}}$ as well
8:        **for** $j = 1, \ldots, r$ **do**
9:           // The training may be replaced with other gradient-based methods as well in practice
10:           $\theta^{(j)} \leftarrow \theta^{(j-1)} - \eta \nabla_\theta \mathcal{L}(\theta^{(j-1)}, \hat{\beta}^{(j-1)}; X_\gamma, \tilde{Y}_\gamma)$
11:           $\hat{\beta}^{(j)} \leftarrow \hat{\beta}^{(j-1)} - \eta \nabla_\beta \mathcal{L}(\theta^{(j-1)}, \hat{\beta}^{(j-1)}; X_\gamma, \tilde{Y}_\gamma)$
12:        // To prevent backpropagation over the GD training
13:        Set $\nabla_\gamma \theta^{(r)} = 0$ and $\nabla_\gamma \hat{\beta}^{(r)} = 0$
14:     Perform estimation

$$\hat{\beta}^{(\infty)} = \hat{\beta}^{(r)} - \begin{bmatrix} 0 \\ J_{p,\beta}^{(r)\top} \end{bmatrix} (\mathcal{J}_\gamma^{(r)} \mathcal{J}_\gamma^{(r)\top})^{-1} \begin{bmatrix} \hat{u}_{\theta^{(r)}}(X_\gamma) - \tilde{Y}_\gamma \\ \mathcal{D}[\hat{u}_{\theta^{(r)}}, \hat{\beta}^{(r)}](X_p) - f(X_p) \end{bmatrix} \qquad (35)$$

15:     **return** $-\|\hat{\beta}^{(\infty)} - \beta_i\|^2$

---

Note that in Line 13 of Alg. 2, we set the gradients $\nabla_\gamma \theta^{(r)}$ and $\nabla_\gamma \hat{\beta}^{(r)}$ to zero. This is done since when implementing the function, it will be possible to write $\theta^{(r)}$ and $\hat{\beta}^{(r)}$ as explicit functions in terms of $\theta$, meaning that when performing back-propagation over the Model Training Estimate criteria, it will also consider these derivatives as well. This can cause memory issues due to the need of back-propagating over many GD steps. Therefore, by explicitly stating that the gradient is zero, it avoids problems during the back-propagation phase. In practice, this can be done via the `stop_gradient` function on JAX, for example.

A speedup that can be applied on MoTE is to instead of performing initial pretraining to obtain the eNTK, we could reuse the NN parameters from the forward PINN in order to compute the eNTK instead. We find that this trick is useful since it gives performance almost as good as performing initial training in each criterion computation, while being more efficient since no additional training needs to be done.

### F.3 TOLERABLE INVERSE PARAMETER CRITERION

### F.3.1 MOTIVATION

To demonstrate the flexibility of PIED, we present another possible method which aims to find the optimal $X$ in (8) without approximating $\hat{\beta}_i$, by taking the opposite approach of choosing $X$ to query noisy observations that would have the least harmful impact when training an inverse PINN already initialized with the right $\hat{\beta}_i$. Intuitively, given an inverse PINN already trained to the correct PDE parameter $\hat{\beta}_i = \beta_i$, a bad choice of $X$ would possibly cause the PDE parameter to drift to other incorrect $\hat{\beta}'$ with further training, while a good choice would retain the correct $\hat{\beta}_i$.

Specifically, during inverse PINN training, the observational data $(X, \tilde{Y}_i)$ influences the loss in (4) explicitly through $\mathcal{L}_{\mathrm{obs}}$, though also implicitly through $\mathcal{L}_{\mathrm{PDE}}$. To see this, note that $(X, \tilde{Y}_i)$ changes $\mathcal{L}_{\mathrm{obs}}$ during training, which adjusts NN parameters $\theta_i$ via gradient descent optimization, including potentially drifting $\hat{\beta}_i$ to a nearby value $\hat{\beta}'$. We could approximate how this shift from $\hat{\beta}_i$ to $\hat{\beta}'$ changes the NN parameters from $\theta_i$ to $\tilde{\theta}_i$ as $\mathcal{L}_{\mathrm{PDE}}(\tilde{\theta}_i(\beta'), \beta')$ is minimized as

$$\tilde{\theta}_i(\beta') \approx \theta_i - \left(\nabla_\theta^2 \mathcal{L}_{\mathrm{PDE}}(\theta_i, \beta_i)\right)^{-1} \left(\nabla_\theta \mathcal{L}_{\mathrm{PDE}}(\theta_i, \beta') - \nabla_\theta \mathcal{L}_{\mathrm{PDE}}(\theta_i, \beta_i)\right), \qquad (36)$$

giving us an approximation of the overall impact of $(X, \tilde{Y}_i)$ on (4) via $\mathcal{L}_{\text{obs}}(\tilde{\theta}_i(\beta'); X, \tilde{Y}_i)$, given by

$$\ell_{X,i}(\beta') \triangleq \mathcal{L}_{\text{obs}}(\tilde{\theta}_i(\beta'); X, \tilde{Y}_i) = \|\hat{u}_{\tilde{\theta}_i(\beta')}(X) - \tilde{Y}_i\|^2. \tag{37}$$

Given a choice of $X$ and its observation values $\tilde{Y}_i$, we could characterize how likely the inverse solver would remain at $\beta_i$ or drift to a neighbouring $\beta'$ after training by considering the Hessian of $\ell_{X,i}(\beta)$ – a "larger" Hessian means a lower chance that $\hat{\beta}_i$ will change as the inverse solver undergoes training to minimize loss, since it is harder for the gradient-based training to "leave" the narrow range of $\beta$. Hence, to optimize for $X$ we could maximize the criterion

$$\hat{\alpha}_{\text{TIP},i}(X) = \log \det \nabla_{\beta'}^2 \ell_{X,i}(\beta_i). \tag{38}$$

### F.3.2 Assumptions and Rough Proof of (36)

We demonstrate the validity of (36), which is adapted from the proof in van der Vaart (2000). For convenience, we will drop the subscript and consider the NN parameters $\theta$ and PDE parameters $\beta$.

Suppose we fix a $\beta$, and let $\theta = \arg\min_{\theta'} \mathcal{L}_{\text{PDE}}(\theta', \beta)$. Since $(\theta, \beta)$ is a minima of $\mathcal{L}_{\text{PDE}}$, we can see that $\nabla_\theta \mathcal{L}_{\text{PDE}}(\theta, \beta) = 0$.

Let $\tilde{\theta}(\beta') = \arg\min_{\theta'} \mathcal{L}_{\text{PDE}}(\theta', \beta')$. Our goal is to approximate $\tilde{\theta}(\beta')$ when $\beta' \approx \beta$.

Let $\Delta\mathcal{L}_{\text{PDE}}(\theta', \beta') = \mathcal{L}_{\text{PDE}}(\theta', \beta') - \mathcal{L}_{\text{PDE}}(\theta', \beta')$ and $\Delta\tilde{\theta}(\beta') = \tilde{\theta}(\beta') - \tilde{\theta}(\beta) = \tilde{\theta}(\beta') - \theta$. We can use this to write

$$\nabla_\theta \mathcal{L}_{\text{PDE}}(\tilde{\theta}(\beta'), \beta') = \nabla_\theta \mathcal{L}_{\text{PDE}}(\theta + \Delta\tilde{\theta}(\beta'), \beta') \tag{39}$$

$$\approx \nabla_\theta \mathcal{L}_{\text{PDE}}(\theta, \beta') + \left(\nabla_\theta^2 \mathcal{L}_{\text{PDE}}(\theta, \beta')\right)\Delta\tilde{\theta}(\beta') \tag{40}$$

$$= \nabla_\theta \mathcal{L}_{\text{PDE}}(\theta, \beta) + \Delta\mathcal{L}_{\text{PDE}}(\theta, \beta') + \left(\nabla_\theta^2 \mathcal{L}_{\text{PDE}}(\theta, \beta')\right)\Delta\tilde{\theta}(\beta') \tag{41}$$

$$= \Delta\mathcal{L}_{\text{PDE}}(\theta, \beta') + \left(\nabla_\theta^2 \mathcal{L}_{\text{PDE}}(\theta, \beta')\right)\Delta\tilde{\theta}(\beta') \tag{42}$$

where (40) arises from performing Taylor expansion on $\nabla_\theta \mathcal{L}_{\text{PDE}}(\tilde{\theta}(\beta'), \beta')$ around $\theta$.

Since $(\tilde{\theta}(\beta'), \beta')$ is a minima of $\mathcal{L}_{\text{PDE}}$, we know that $\nabla_\theta \mathcal{L}_{\text{PDE}}(\tilde{\theta}(\beta'), \beta') = 0$, and therefore we can solve for $\Delta\tilde{\theta}(\beta')$ to obtain

$$\Delta\tilde{\theta}(\beta') \approx -\left(\nabla_\theta^2 \mathcal{L}_{\text{PDE}}(\theta, \beta')\right)^{-1} \Delta\mathcal{L}_{\text{PDE}}(\theta, \beta') \tag{43}$$

which can be rewritten to match the form in (36).

Some readers may question whether (36) is valid for NNs where the learned NN parameter may not be a global minima. We note that despite this, we are only interested in the curvature around a minima anyway, and so we can still inspect the change of that minima as the loss function changes regardless. Furthermore, this technique has been used for NNs in other applications as well, one notable instance being the influence function (Koh & Liang, 2017) which aims to study how the test performance of supervised learning tasks changes as certain training examples are upweighed. In the paper, they are able to design a scoring function based on the same mathematical tool and successfully interpret performances of NNs. In our work, we find that despite the assumptions on the global minima is not met, we are still able to achieve good empirical results as well.

### F.3.3 Choice of Loss Function Used in (36)

Note that in (36), we compute the Hessian w.r.t. the forward PINN loss. Some readers may wonder why the overall PINN loss from (4) is not used instead.

This choice is due to two main reasons. First, it is more computationally efficient. Notice that the change in NN parameter in (36) depends on $\mathcal{L}_{\text{PDE}}$, and therefore are independent of the observations and hence independent of the design parameters $\gamma$. This leads to the optimizing of the final criterion to not require differentiating (36) with respect to $\gamma$, reducing the computational load during optimization. We find that doing so does not cause significant effect in the obtained design parameter $\gamma^*$.

Second, this matches more closely to the inverse problem setup as described in Sec. 2. In the IP as described, the objective (2) is usually to find the $\beta$ whose output matches that of $Y$. In TIP, the

tolerable parameters is the Hessian based on the objective (37), which can also be interpreted in a similar way as (2). Furthermore, through this interpretation, TIP also exhibits a stronger connection to Bayesian methods, as discussed in App. F.3.4.

### F.3.4 BAYESIAN INTERPRETATION OF TIP

Interestingly, TIP can be seen as the application of the Laplace approximation on the posterior distribution of $\beta$ after observing data $(X_\gamma, Y)$, assuming a uniform prior and Gaussian observation noise, and view the criterion score $\hat{\alpha}_{\text{TIP},i}$ as the information gain of $\hat{\beta}_i$ given observational data $(X_\gamma, \tilde{Y}_{\gamma,i})$. Assume that the observed data is generated from the true underlying function with added Gaussian noise. Then, the likelihood function can be written as

$$p(Y|\beta, X_\gamma) = \mathcal{N}(Y|u_\beta(X_\gamma), \sigma^2 I) = \prod_{j=1}^{M} \mathcal{N}(y_j|u_\beta(x_{\gamma,j}), \sigma^2). \tag{44}$$

If we assume a uniform prior over $\mathcal{B}$ (i.e., assume $p(\beta) = c$ for some constant $c$), then it is simple to show that $p(\beta|X_\gamma, Y) = p(Y|\beta, X_\gamma)/p(Y|X_\gamma)$, where $p(Y|X_\gamma)$ can be treated as a constant. In this case, we can write the log posterior as

$$\log p(\beta|X_\gamma, Y) = \left[ \sum_{j=1}^{N} \log \mathcal{N}(y_j|u_\beta(x_{\gamma,j}), \sigma^2) \right] - \log p(Y|X_\gamma) \tag{45}$$

$$= \sum_{j=1}^{N} \left[ \frac{1}{\sigma\sqrt{2\pi}} - \frac{(u_\beta(x_{\gamma,j}) - y_j)^2}{2\sigma^2} \right] - \log p(Y|X_\gamma) \tag{46}$$

$$= -\frac{\|u_\beta(X_\gamma) - Y\|^2}{2\sigma^2} + \text{constant.} \tag{47}$$

To make (47) tractable, we can apply Laplace's approximation on the posterior. To do so, we perform a Taylor expansion on $\log p(\beta|X_\gamma, Y)$ around the MAP of the distribution. In this case, we would expect the MAP to be at the true PDE parameter $\beta_0$, where $\frac{\partial}{\partial \beta} \log p(\beta|X_\gamma, Y) = 0$. Once expanded, this would give

$$\log p(\beta|X_\gamma, Y) \approx \log p(\beta_0|X_\gamma, Y) + (\beta - \beta_0)^\top \left[ \nabla_\beta^2 \log p(\beta_0|X_\gamma, Y) \right] (\beta - \beta_0) \tag{48}$$

$$= \log p(\beta_0|X_\gamma, Y) - \frac{1}{2\sigma^2} (\beta - \beta_0)^\top \left[ \nabla_\beta^2 \frac{\|u_{\beta_0}(X_\gamma) - Y\|^2}{2\sigma^2} \right] (\beta - \beta_0). \tag{49}$$

Note that this can also be written as

$$p(\beta|X_\gamma, Y) \approx p(\beta_0|X_\gamma, Y) \exp\left( -(\beta - \beta_0)^\top \left[ \nabla_\beta^2 \frac{\|u_{\beta_0}(X_\gamma) - Y\|^2}{2\sigma^2} \right] (\beta - \beta_0) \right) \tag{50}$$

which confirms that the Taylor expansion approximates the posterior distribution as a Gaussian distribution with mean $\mu_{\text{Laplace}} = \beta_0$ and covariance matrix $\Sigma_{\text{Laplace}} = \nabla_\beta^2 \frac{\|u_{\beta_0}(X_\gamma) - Y\|^2}{2\sigma^2}$. Since the posterior is approximated as a multivariate Gaussian distribution, it is simple to approximate the entropy of $\beta$ as distributed by $p(\beta|X_\gamma, Y)$ using the entropy of multivariate Gaussian distribution as

$$\mathbb{H}[\beta|X_\gamma, Y] \approx \frac{1}{2} \log \det \Sigma_{\text{Laplace}} + \frac{M}{2} \log 2\pi e \tag{51}$$

$$= \frac{1}{2} \log \det \nabla_\beta^2 \|u_{\beta_0}(X_\gamma) - Y\|^2 + \text{constant.} \tag{52}$$

Finally, from (11), we can approximate the EIG as

$$\text{EIG}(\gamma) = -\mathbb{E}_{Y' \sim p(Y|X_\gamma)} \left[ \mathbb{H}[\beta|X_\gamma, Y = Y'] \right] \tag{53}$$

$$= \mathbb{H}[\beta] - \mathbb{E}_{\beta_0 \sim p(\beta), Y' \sim p(Y|\beta_0, X_\gamma)} \left[ \mathbb{H}[\beta|X_\gamma, Y = Y'] \right] \tag{54}$$

$$\approx \mathbb{H}[\beta] - \frac{1}{2} \mathbb{E}_{\beta_0 \sim p(\beta), Y' \sim p(Y|\beta_0, X_\gamma)} \left[ \log \det \nabla_\beta^2 \|u_{\beta_0}(X_\gamma) - Y'\|^2 \right] + \text{constant} \tag{55}$$

where (55) uses the approximation of entropy in (52). Note that $\mathbb{H}[\beta]$ is a constant independent of $\gamma$ and therefore can be ignored.

Note that in the derivation so far, we have assumed that we are able to compute the PDE solution $u_\beta$ *and* be able to compute how the solution output changes w.r.t. $\beta$. This, fortunately, is made possible using PINNs. Specifically, in the likelihood distribution $p(Y|\beta, X_\gamma)$ from (44) and as sampled from in the expectation in (55), we can replace the $u_\beta$ with a NN $\hat{u}_{\tilde{\theta}(\beta)}$ with parameter $\tilde{\theta}(\beta)$ as defined in (36). Similarly, inside the expectation term of (55), we can replace $u_{\beta_0}(X_\gamma)$ with $\hat{u}_{\tilde{\theta}(\beta_0)}$, and $Y'$ with a noisy reading of $\hat{u}_{\tilde{\theta}(\beta_0)}$. Ultimately, ignoring additive constants, this gives

$$\text{EIG}(\gamma) \approx -\frac{1}{2}\mathbb{E}_{\beta_0 \sim p(\beta), \varepsilon \sim \mathcal{N}(\varepsilon|0, \sigma^2 I)}\left[\log\det\nabla_\beta^2\left[\|\hat{u}_{\tilde{\theta}(\beta)}(X_\gamma) - \hat{u}_{\tilde{\theta}(\beta_0)}(X_\gamma) + \varepsilon\|^2\right]_{\beta=\beta_0}\right] \quad (56)$$

For simplicity, we can ignore the additive Gaussian noise in the approximation of $Y$, i.e., ignore the $\varepsilon$ term, and the term inside the expectation is the same as that in (38).

This analysis links TIP to some existing ED methods for IPs based on Laplace's approximation (Beck et al., 2018; Alexanderian et al., 2024). Despite these links, our method remains novel in that through the use of PINNs, we can consider the Hessian of the PDE solution directly, allowing the resulting criterion to be differentiable w.r.t. $\gamma$. This is unlike past works which often require more careful analysis of the specific PDE involved, and relies on discretized simulations (and therefore are not differentiable w.r.t. the input points). We note that while previous works have proposed the use of the Hessian of the learned inverse posterior distribution (Beck et al., 2018; Alexanderian et al., 2024), our method is novel in that it considers the Hessian (and hence the sensitivity) of the PDE solution directly, rather than of a posterior distribution. This is due to the usage of PINNs and its differentiability in the ED process directly, which has not been done in past works.

We comment about the assumptions required to arrive at the approximation in (56).

- We assume that the data is generated with random Gaussian noise. This is a standard assumption as done in other IP and ED methods in the literature.
- We assume that the posterior distribution is unimodal. Note that this assumption does not always hold. One example where this assumption does not hold would be in the case where multiple values of $\beta$ may represent the same PDE parameter (e.g., $\beta$ represents NN parameterization of an inverse function). In our experiments, we find that the performance of the method remains good regardless. Furthermore, we believe that the issue can be mitigated by considering the problem under some embedding space $\phi(\beta)$ where two embeddings are similar when their parameterizations represent similar functions. This is likely possible by adjusting our criterion to incorporate $\phi$ through Lagrange inversion theorem, however we will defer this point to a future work.

  Another case where this assumption may not hold is when there are some degeneracy in the inverse problem solution. In this case, the problem cannot be alleviated anyway unless more observations data are acquired (a simple way to think about this is when there is only one observation reading is allowed, and therefore a good PDE parameter solution will not be obtainable regardless of the ED method).
- We assume that the unimodal distribution is maximal at $\beta_0$. Given that the distribution is unimodal, this point would likely hold since the pseudo-observation from the NN is already obtained from using PDE parameter of $\beta_0$.

### F.3.5 APPROXIMATION OF HESSIAN IN (36)

Instead of computing the Hessian of $\mathcal{L}_{\text{PDE}}$ directly, we can also employ a trick which avoids computing the Hessian directly, but instead approximates the Hessian based on the first-order derivatives.

Suppose we define

$$R(\theta, \beta) = \begin{bmatrix} |X_p|^{-1/2}\big(\mathcal{D}[\hat{u}_\theta, \beta](X_p) - f(X_p)\big) \\ |X_b|^{-1/2}\big(\mathcal{B}[\hat{u}_\theta, \beta](X_b) - g(X_b)\big) \end{bmatrix}. \quad (57)$$

We can see that

$$\mathcal{L}_{\text{PDE}}(\theta, \beta) = \frac{1}{2}R(\theta, \beta)^\top R(\theta, \beta). \quad (58)$$

We can then write

$$\nabla_\theta \mathcal{L}_{\text{PDE}}(\theta, \beta) = \nabla_\theta R(\theta, \beta)^\top R(\theta, \beta) \tag{59}$$

and

$$\nabla_\theta^2 \mathcal{L}_{\text{PDE}}(\theta, \beta) = \nabla_\theta R(\theta, \beta)^\top \nabla_\theta R(\theta, \beta) + \nabla_\theta^2 R(\theta, \beta)^\top R(\theta, \beta). \tag{60}$$

In the case that $(\theta, \beta)$ are obtained after PINN training has converged, we would have $R(\theta, \beta) \approx 0$, which means we can write

$$\nabla_\theta^2 \mathcal{L}_{\text{PDE}}(\theta, \beta) \approx \nabla_\theta R(\theta, \beta)^\top \nabla_\theta R(\theta, \beta). \tag{61}$$

We therefore can approximate the Hessian as used in (36) using first-order derivatives instead.

### F.3.6 Comparison with Other Benchmarks

Table 3 shows the results of TIP compared to our other methods. We see that while in some cases we are able to get comparable results to FIST or MoTE, it often will perform worse than these other methods. This is likely due to the stronger assumptions that are required for the method.

Table 3: Results for the inverse problems using TIP criterion compared to our other proposed criteria. The table is interpreted similarly to Table 1.

| Dataset | Finite-dimensional | | Function-valued | Real dataset | |
|---|---|---|---|---|---|
| | Wave ($\times 10^{-1}$) | Navier-Stokes ($\times 10^{-2}$) | Eikonal ($\times 10^1$) | Groundwater ($\times 10^1$) | Cell Growth ($\times 10^0$) |
| FIST | 3.87 (0.76) | 2.10 (1.45) | 0.74 (0.02) | 1.93 (0.08) | 2.62 (0.11) |
| MoTE | 3.81 (2.34) | 1.18 (0.11) | 0.76 (0.02) | 2.00 (0.60) | 2.83 (0.04) |
| TIP | 5.11 (0.01) | 9.04 (2.04) | 0.75 (0.01) | 2.25 (0.26) | 2.94 (0.23) |

## G Complete Algorithm For PIED

We summarize PIED via a pseudocode presented in Alg. 3. The ED procedure consists of three main phases – learning of the shared PINN parameter initialization, the criterion generation phase which consists of performing forward simulations and consequently defining the criteria, and the criterion optimization phase which proceeds to perform constrained continuous optimization on the criterion.

Note that in our framework, the forward simulation only has to be ran once per ED loop, and can all be ran in parallel using packages which allows for parallelization such as `vmap` on JAX. We also find that the forward simulation can be used without explicitly injecting artificial noise, while still giving observation inputs which work well for IPs involving noisy data.

In the criterion optimization phase, we use projected gradient descent to ensure the resulting design parameter is in the bounded space. However, other constrained optimization algorithms could be used as well, e.g., L-BFGS-B. We repeat the optimization loop over many runs due to the potential non-convexity of the criteria, to obtain a better estimate of the optima.

## H Details About The Experimental Setup

### H.1 General ED Loop and IP Setup

Our experiment consists of two phases. In the first phase, we perform the ED loop using PIED or with the other benchmarks. Here, we allow a fixed number of forward simulations using PINNs, which the ED methods can query from as many times as it wants. Each ED methods have time restrictions, where they are allowed to run either for a certain duration, or until they have completed some fixed number of iterations.

After the ED methods have selected the optimal design parameters, the same design parameters are used to test on multiple instances of the IP (we run at least 10 of such instances depending on how much computation resources the specific problem requires). In each instance of the IP, we draw a random ground-truth PDE parameter, and generate the observations according to the model and the random ground-truth PDE parameter. The IP is solved using inverse PINNs to obtain a guess of the

---

**Algorithm 3** PIED

---

    // Learning shared NN parameters
1: Randomly initialize $\theta_{\mathrm{SI}}$
2: **for** $s$ rounds **do**
3:     Randomly sample $\beta'_1, \ldots, \beta'_k$
4:     **for** $j = 1, \ldots, k$ **do**
5:         $\theta'_j \leftarrow$ NN parameter after training $\theta_{\mathrm{SI}}$ with training loss $\mathcal{L}_{\mathrm{PDE}}(\theta, \beta'_j)$
6:     $\theta_{\mathrm{SI}} \leftarrow \frac{1}{k} \sum_{j=1}^k \theta'_j$     ▷ Used as initialization for all proceeding forward and inverse PINNs
    // Criterion generation phase
7: **for** $i = 1, \ldots, M$ **do**
8:     Randomize PDE parameter $\beta_i$
9:     $\tilde{u}_{\beta_i} \leftarrow \mathbf{F}(\beta_i)$                                 ▷ Forward simulation
10:     **if** use FIST criterion **then**
11:         Define $\hat{\alpha}_i$ to FIST criterion from Alg. 1
12:     **else if** use MoTE criterion **then**
13:         Define $\hat{\alpha}_i$ to MoTE criterion from Alg. 2
14: Define aggregated criterion $\alpha(X_\gamma) = \frac{1}{N} \sum_{i=1}^N \hat{\alpha}_i(X_\gamma)$
    // Criterion optimization phase
15: Initialize $\gamma_{\mathrm{best}} \in \mathcal{S}_\gamma$ randomly
16: **repeat**
17:     Initialize $\gamma' \in \mathcal{S}_\gamma$ randomly
18:     **for** $p$ training steps **do**                       ▷ Gradient-based optimization
19:         $\tilde{\gamma}' \leftarrow \gamma' + \eta \nabla_\gamma \alpha(X_{\gamma'})$     ▷ Gradient ascent since the criterion should be maximized
20:         $\gamma' \leftarrow \mathrm{proj}_{\mathcal{S}_\gamma}(\tilde{\gamma}')$                   ▷ Perform projection s.t. $\gamma' \in \mathcal{S}_\gamma$
21:     **if** $\alpha(X_{\gamma'}) > \alpha(X_{\gamma_{\mathrm{best}}})$ **then**
22:         $\gamma_{\mathrm{best}} \leftarrow \gamma'$
23: **until** computational limit hit
24: **return** $\gamma_{\mathrm{best}}$

---

Table 4: Architectures of used NNs

| Problem | Depth | Width | Activation | Output transformation |
|---|---|---|---|---|
| Damped oscillator | 6 | 8 | `tanh` | None |
| 1D wave | 3 | 16 | `sin` | None |
| 2D Navier-Stokes | 6 | 16 | `sin` | None |
| 2D Eikonal (modelling $u_\beta$) | 6 | 8 | `tanh` | $(x, y) \to y\|x - x_0\|$ |
| 2D Eikonal (modelling $\beta$) | 1 | 16 | `sin` | $(x, y) \to |y| + 0.2$ |
| Groundwater flow | 2 | 8 | `tanh` | None |
| Cell population | 2 | 8 | `tanh` | None |

PDE parameter. For each instance, we can obtain an error score, which measures how different the PDE parameter estimate is from the ground-truth value.

For each problem, we repeat the ED and IP loop five times, where in each time we obtain multiple values for the IP error. In our results, we report the distribution of all the error scores obtained through the percentile values of the error (i.e., $p$ percent of all IP instances using a certain ED methods have errors of at most $x$), removing some of the extreme values (in the main paper, we remove the top and bottom ten percent, while in the Appendix we show the distribution via a boxplot removing the outliers). This is done since some PDE parameters result in IPs which are easier than others, and to demonstrate the performance of each ED methods across all possible PDE parameters.

### H.2 PINN AND PINN TRAINING HYPERPARAMETERS

The architectures of the PINNs and other NNs used are listed in Table 4. Note that we only use multi-layer perceptrons in our experiments. The training process hyperparameters for the forward and inverse PINNs are listed in Table 5.

Table 5: Training hyperparameters

| Problem | Training steps | # PDE Col. Pts. | # IC/BC Col. Pts. | Optimizer |
|---|---|---|---|---|
| Damped oscillator | 30k | 300 | 1 | Adam (lr = 0.01) |
| 1D wave | 200k | 15k | 2k | L-BFGS |
| 2D Navier-Stokes | 100k | 2k | 300 | Adam (lr = 0.001) |
| 2D Eikonal | 50k | 10k | 1 | Adam (lr = 0.001) |
| Groundwater flow | 50k | 500 | 1 | L-BFGS |
| Cell population | 50k | 1k | 100 | Adam (lr = 0.001) |

### H.3 Scoring Metric

To judge how well our ED methods perform, we will use the error $L(\hat{\beta}, \beta)$ of the PDE parameter $\beta$. When $\beta$ has finite dimensions (i.e., represented as a scalar value or as a vector value), then the loss is simply the MSE loss, i.e.,

$$L(\hat{\beta}, \beta) = \left\| \hat{\beta} - \beta \right\|_2^2. \tag{62}$$

For the case where $\beta$ is a function, we select some number of test points $\{x_{T,i}\}_{i=1}^{N_{\text{test}}}$ and compute the MSE loss of the estimated function on those test points, i.e.

$$L(\hat{\beta}, \beta) = \sum_{i=1}^{N_{\text{test}}} \left\| \hat{\beta}(x_{T,i}) - \beta(x_{T,i}) \right\|_2^2. \tag{63}$$

### H.4 Benchmarks for the Scoring Criterion

In this section we describe a few scoring criteria we use as a benchmark. We first describe the benchmarks which are based on methods of estimating the expected information gain (EIG).

- **Mutual Information Neural Estimator (MINE) (Belghazi et al., 2018).** The estimator utilizes Donsker-Varadhan representation of the KL Divergence to show that we can provide a lower bound to the EIG as

$$\text{EIG}(X_\gamma) \geq \mathcal{L}_{\text{DV}}(X_\gamma) \tag{64}$$
$$\triangleq \mathbb{E}_{(y,\beta) \sim p(\beta)p(Y|\beta,X_\gamma)} \left[ T_\phi(Y, \beta|X_\gamma) \right] - \log \mathbb{E}_{(Y,\beta) \sim p(\beta)p(y|X_\gamma)} \left[ e^{T_\phi(Y,\beta|X_\gamma)} \right] \tag{65}$$

  where $T_\phi$ is a parametrized family of functions.

  Note that while the estimator $\mathcal{L}_{\text{DV}}(X_\gamma)$ relies on sampling $p(Y|\beta, X_\gamma)$ and $p(Y|X_\gamma)$ directly, this would require running many forward simulations for different $\beta$ samples. Instead, to sample from these two distributions, we draw $M$ random samples of $\beta$ and approximate $p(\beta)$ with a mixture of Dirac-delta distributions, i.e.,

$$p(\beta) \approx \hat{p}(\beta) \triangleq \frac{1}{M} \sum_{i=1}^{M} \delta(\beta - \beta_i) \quad \text{where} \quad \beta_1, \ldots, \beta_M \sim p(\beta). \tag{66}$$

  In this case, the forward simulation only needs to be ran for $M$ samples of $\beta$ and the distributions $p(Y|\beta, X_\gamma)$ and $p(Y|X_\gamma)$ can be efficiently approximated.

- **Variational Bayesian Optimal ED (VBOED) estimator (Foster et al., 2019).** The original paper desicribes multiple estimators for the EIG, however we will use the variational marginal estimator. The estimator utilizes the fact that we can provide an upper bound to the EIG as

$$\text{EIG}(X_\gamma) \leq \mathcal{U}_{\text{marg}}(X_\gamma) \triangleq \mathbb{E}_{(Y,\beta) \sim p(\beta)p(Y|\beta,X_\gamma)} \left[ \log \frac{p(Y|\beta, X_\gamma)}{q_\phi(Y|X_\gamma)} \right] \tag{67}$$

  where $q_\phi$ is a variational family parametrized by $\phi$. To compute the EIG, we find the $\phi$ which minimizes $\mathcal{U}_{\text{marg}}(X_\gamma)$. Instead of computing the upper bound exactly, we use an empirical estimation based on samples of $(Y, \beta)$ generated from the PINNs with added noise. Similar

Table 6: Hyperparameters used for different criteria in PIED. Note that for MoTE, the case when $r = \infty$ refers to when we use the forward PINN for the NTK regression step.

| Dataset | FIST | | MoTE |
|---|---|---|---|
| | $\sigma_p^2$ | $r$ | $r$ |
| Damped oscillator | 0.5 | 50 | $\infty$ |
| 1D wave | 0.5 | 200 | 0 |
| 2D Navier-Stokes | 0.5 | 100 | $\infty$ |
| 2D Eikonal | 0.01 | 200 | 0 |
| Groundwater flow | 0.01 | 100 | $\infty$ |
| Cell population | 0.1 | 100 | 1000 |

to MINE, we approximate $p(\beta)$ with a mixture of Dirac-delta distributions (66) such that only a limited number PINN forward simulations are required.

Note that while Foster et al. (2019) does propose a variational NMC (VNMC) estimator as well, this requires computing $p(Y, \beta | X_\gamma) = p(\beta)p(Y|\beta, X_\gamma)$ for a randomly sampled $\beta$. It is not feasible to compute $p(Y|\beta, X_\gamma)$ on the fly since this would require running a costly forward simulation for the randomly sampled $\beta$, and therefore the method is not included for this benchmark.

We also use other benchmarks which are not based on the EIG, listed as follows.

- **Random.** The design parameters are chosen randomly.

- **Grid.** The design parameters are chosen such that the sensor readings are placed regularly in some fashion. For the 1D examples, the sensors are placed such that they all regularly spaced out. For the 2D examples, the sensors are placed such that they are shaped in a regular 2D grid with each sides having as equal number of sensors as possible. Note that no optimization is done, but instead the observation input configuration is fixed per problem.

- **Mutual information (MI) (Krause et al., 2008).** The criterion considers the outputs $Y_{X_\gamma}$ of the chosen observation input $X_\gamma$ and the outputs $Y_{X_t}$ of some test set $X_t$, and defines the score to be the mutual information between $Y_{X_\gamma}$ and $Y_{X_t}$, i.e.,

$$\alpha_{\mathrm{MI}}(X_\gamma) = \mathrm{MI}(Y_{X_t \setminus X_\gamma}; Y_{X_\gamma}) = \mathbb{H}[Y_{X_t}] - \mathbb{H}[Y_{X_t \setminus X_\gamma} | Y_{X_\gamma}]. \tag{68}$$

In our experiments, we approximate the observation outputs via a Gaussian process (GP) whose kernel is the covariance of the PDE solutions, i.e., $K(x, x') = \mathrm{Cov}_{\beta \sim p(\beta)}[u_\beta(x), u_\beta(x')]$, where we approximate the covariance using the forward simulations. By approximating the output using a GP, the entropies $\mathbb{H}[Y_{X_t \setminus X_\gamma}]$ and $\mathbb{H}[Y_{X_t \setminus X_\gamma} | Y_{X_\gamma}]$ can be written directly in terms of the approximate kernel function. Also, since we do not perform discretization and treat the problem as a combinatorial optimization one due to the additional point constraints, we chose to let $X_t \setminus X_\gamma = X_t$ for simplicity.

We also run the two criteria proposed as the benchmark, listed below.

- **Few-step Inverse Solver Training (Alg. 1).** For each of the trial, we note the value of parameter perturbation $\sigma_p^2$ and training steps $r$ used.

- **Model Training Estimate (Alg. 2).** For each trial, we note how many initial training steps $r$ are used, or if we just re-use the NN parameters from the forward PINN for the eNTK.

In Table 6, we show the hyperparameters for the criteria we use for the main results.

In the results, we add "$+\mathtt{SI}$" suffix to indicate the benchmark where the shared NN initialization is used for all of the forward and inverse PINNs involved during ED and IP phases. MINE, VBOED, NMC and MI are optimized using Bayesian optimization (Frazier, 2018).

### H.5    IMPLEMENTATION AND HARDWARE

All of the code were implemented based on the JAX library (Bradbury et al., 2018), which allows for NN training and auto-differentiation of many mathematical modules within. Criteria which are

optimized by Bayesian optimization are done so using BOTORCH (Balandat et al., 2020), while criteria optimized using gradient-based methods are done so using JAXOPT (Blondel et al., 2021).

The damped oscillator, wave equation, Eikonal equation and groundwater experiments were conducted on a machine with AMD EPYC 7713 64-Core Processor and NVIDIA A100-SXM4-40GB GPU, while the remaining experiments were done on AMD EPYC 7763 64-Core Processor CPU and NVIDIA L40 GPU.

# I  ADDITIONAL EXPERIMENTAL RESULTS

## I.1  ADDITIONAL RESULTS FROM EXPERIMENTS ON LEARNED NN INITIALIZATION

In Fig. 9, we present examples of the learned NN initialization and also its performance for forward PINNs for the 2D Eikonal equation example. We see that this shows similar trends to the examples from before as shown in Fig. 2.

In Fig. 10, we show the test error for an individual forward PINN when performing forward simulation for a value of $\beta$, when using and not using a learned NN initialization. We see that when using a learned NN initialization, the test loss typically is already lower than that from random initialization at the start, and also tends towards convergence much faster. Even when the test loss is higher at the start, it is able to catch up to the performance of the randomly initialized PINN under much fewer training steps.

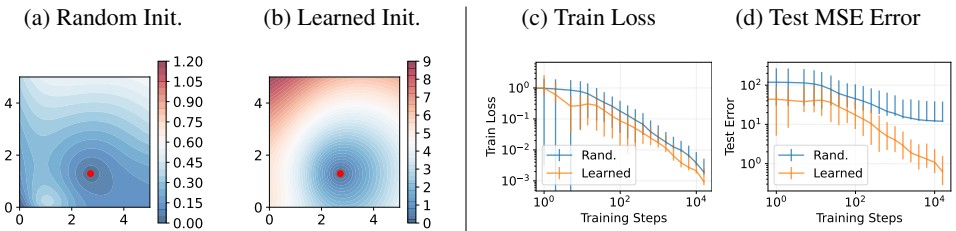

Figure 9: Results for learning a NN initialization for PINNs trained on 2D Eikonal equation case. The interpretation is the same for that in Fig. 2.

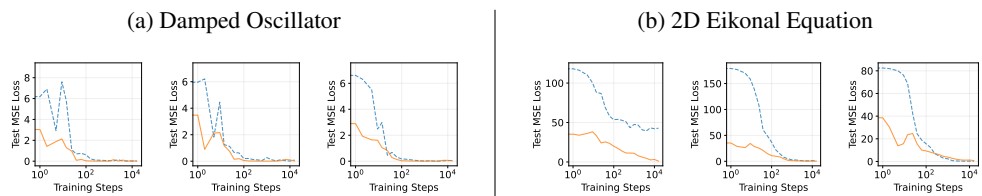

Figure 10: Examples of test error of forward PINNs for each problems for different values of $\beta$. Each plot represents the PINN training for one random random value of $\beta$. The dotted blue lines represent when the PINN is initialized with a random NN parameters, while the solid yellow lines represent when the PINN is initialized form the learned initialization.

## I.2  TEST ON DISTRIBUTION MISMATCH

In Fig. 11, we present the results for the damped oscillator experiments for when the prior distribution during the ED process and for the tested IPs are different. In the distribution mismatch case, we use PDE parameter range $\mu, k \in [0, 2]$ during the ED process, but use the values $\mu, k \in [0, 4]$ for the true ground truth value in the IP process. From the results, we see that our benchmarks are still able to retain good performances over the benchmarks even when the prior distribution of the PDE parameters are misspecified.

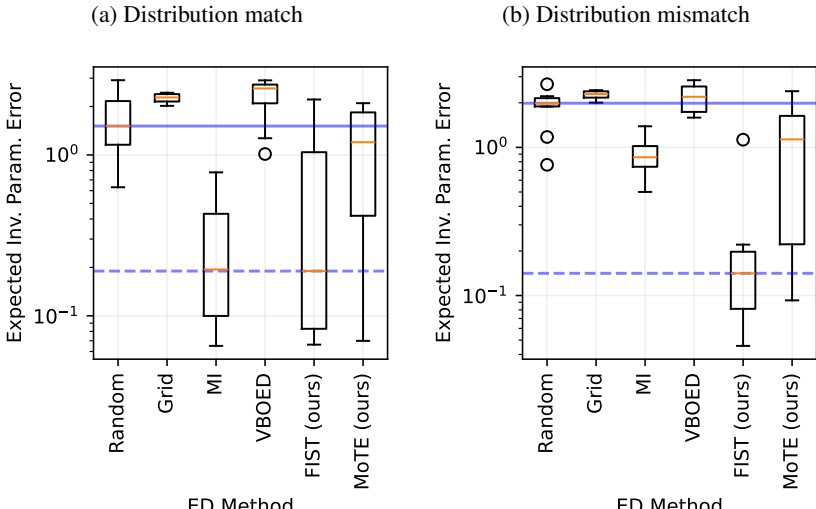

Figure 11: Results for the 1D damped oscillator example for distribution match and mismatch. In both cases, the thick blue line represents the performance of Random baseline, while the dashed line represents the best performing benchmark.

## J    DISCUSSIONS ON GENERAL LIMITATIONS AND SOCIETAL IMPACTS

In our experiments, we have mostly conducted experiments based on vanilla PINN architectures. We have not done verification of whether PIED works well for other more complex architectures such as physics-informed deep operators. Future work on this area would be interesting.

The work also relies on the use of PINNs as the forward simulators and IP solvers, and are not applicable to other forward simulators or IP solvers. While PINNs are well-suited for both tasks, they also still pose practical problems such as difficulty in training for certain problems. The problem can be mitigated through more careful selection of collocation points (Wu et al., 2023; Lau et al., 2024a), which will be interesting to consider in future works to further boost the performance of PIED.

We believe the work has minimal negative and significant positive societal impacts, since they can be used in many science and engineering applications where costs of data collection from experiments can be prohibitive. This means that the cost barrier in performing effective scientific experiments can be lowered allowing for further scientific discoveries. We note that our work could potentially be applied for a range of scientific research, which may include unethical or harmful research done by malicious actors. However, this risk applies to all tools that accelerate scientific progress, and we believe that existing policies and measures guarding against these risks are sufficient.

