# OpenReview forum: "PIED: Physics-Informed Experimental Design for Inverse Problems"
_ICLR.cc/2025/Conference — ICLR 2025 Poster_

### Official Review · Reviewer_CY8h · 2024-11-01

**Soundness:** 3
**Presentation:** 3
**Contribution:** 2
**Rating:** 6
**Confidence:** 3

**Summary:**

This paper solves the one-shot experiment design using PINNs in both forward and inverse problems. It overcomes computational bottlenecks by parallelism and meta-learning of initialization. The experiments on both synthetic and real-life datasets show the performance improvements.

**Strengths:**

1. this paper joins many advanced techniques together to solve the problem.
2. most of the representation is clear, with extensive experiments and details.

**Weaknesses:**

1. The overall algorithm flow is unclear in the main text. Figure 1b is too general and missing almost all of the details of the technique.
2. Does not compare with existing NN-based experiment design methods.
3. Almost all techniques are adopted from existing literature.

**Questions:**

1. What's the relationship between the 'PINN for ED' and the 'adaptive-sampling PINN for inverse problem'?
2. Can a trained PINN generalize to a new ED problem with new $\beta$? Can the model (including the meta-learning part) generalize to a new distribution of $\beta$？

---

> ### Author Response · Authors · 2024-11-21
> **Response to Reviewer CY8h (Part 1/2)**
>
> We would like to thank the reviewer for taking the time to review our work, and for recognizing that our work "joins many advanced techniques together" and has "extensive experiments and details". We have provided clarifications to your questions and concerns below.
>
> ---
>
> > The overall algorithm flow is unclear in the main text. Figure 1b is too general and missing almost all of the details of the technique.
>
> We thank the reviewer for the feedback. We had originally intended for Fig. 1b in Sec.3 to mainly provide an overview of the different components in our PIED framework, with a comparison on how it mirrors the components required for solving the inverse problem during inference time (Fig. 1a). We then built on this by describing the training and optimization methods used in the implementation of PIED later in Sec. 4, with details including an overview and detailed pseudocode of PIED (Alg. 3) in Appendix G due to space limitations. However, we agree that we could be more explicit in describing the full PIED framework implementation in the main paper, and will adjust accordingly.
>
> ---
>
> > Does not compare with existing NN-based experiment design methods.
>
> We thank the reviewer for this comment. We find that while many works have been proposed to use NNs to amortize the cost of inferring the unknown inverse parameter, few have focused directly on using these NNs for ED methods. Nonetheless, we have ran additional experiments on two benchmarks which uses NNs in order to infer the optimal observation parameters:
>
> - Neural likelihood estimation (NLE) from [1], where a NN is used to learn the likelihood function of an observation given the inverse parameters and observation inputs. In this scenario, we perform ED by using the learned likelihood function to obtain an estimate of the mutual information.
>
> - Mutual Information Neural Estimation (MINE) from [2], where a NN is used to compute a lower bound for mutual information computation and therefore used as the ED criterion.
>
> We include some additional results for both of these methods in the table below. From the table, we see that PIED is still able to outperform these baselines while also being more efficient in terms of computation time to arrive at those configurations.
>
> |             | Eikonal ($\times 10^{1}$)   | Groundwater ($\times 10^{1}$)   | Cell Growth ($\times 10^{0}$)   |
> |:------------|:----------------------------|:--------------------------------|:--------------------------------|
> | MINE        | 2.43 (0.24)                 | 2.21 (2.16)                     | 3.50 (1.44)                     |
> | NLE         | 1.63 (0.22)                 | 2.27 (1.59)                     | 3.65 (1.10)                     |
> | FIST (ours) | **0.74 (0.05)**        | **1.93 (0.27)**           | **2.62 (0.36)**           |
> | MoTE (ours) | **0.76 (0.04)**        | **2.00 (0.95)**            | 2.83 (0.27)                     |
>
> [1] Papamakarios et al. Sequential Neural Likelihood: Fast Likelihood-free Inference with Autoregressive Flows.
>
> [2] Belghazi et al. Mutual Information Neural Estimation.
>
> ---
>
> > Almost all techniques are adopted from existing literature.
>
> We would like to emphasize that while at a high level many of the various techniques used in PIED may seem similar to that in existing literature, in this regard our work's technical novelty and contribution lies in identifying the appropriate techniques required to effectively solve the various technical challenges in our novel problem setting, and also how they can be integrated well in an overall framework to yield significant empirical performance advantages, both of which we believe are highly non-trivial. We highlight some examples here:
> - We proposed a novel ED framework comprising PINNs that serve as both forward simulators and inverse solvers with the same model architecture, allowing us to utilize the efficiency advantages of PINNs and enable gradient-based methods for the continuous optimization of the experimental design parameters.
> - We introduced the use of meta-learning over different reference PDE parameters $\beta$ values such that the shared PINN initialization could be used for speeding up both forward simulations, and also inverse solvers for various different $\beta$ values concurrently, which has not been done in past works.
> - While the use of NTK in PINNs is not new, our method (MoTE) using NTK regression on the NN to estimate the output PDE parameters $\beta$ is new and has not been done before in past works. This is in addition to techniques used to better account for evolving NTKs due to the finite-width of the PINNs.
> - Our FIST method is also uses novel tricks such as analyzing effectiveness of an observation input based on the perturbed NN parameters corresponding to $\beta$, partially training the inverse PINNs, and back-propagating through the inverse PINNs' training steps to optimize for observation points -- this is also not done before in past works.

---

> ### Author Response · Authors · 2024-11-21
> **Response to Reviewer CY8h (Part 2/2)**
>
> > What's the relationship between the 'PINN for ED' and the 'adaptive-sampling PINN for inverse problem'?
>
> We would like to clarify that on their own, PINNs cannot be used for experimental design (ED) problems, where the goal is to optimize for observation points that yield the most informative measurements for estimating the unknown inverse parameters. Our PIED framework makes use of PINNs, which can be used both as forward simulators and inverse solvers, to solve the ED problem efficiently given the various advantages of PINNs as outlined in our main paper.
>
> Our framework also focuses on the non-adaptive setting (Lines 127-134, Sec. 2) where the observation points need to be selected up-front for one-shot deployment of sensors. This is unlike the adaptive setting, where multiple rounds of observations are allowed and the design parameter can be adjusted between each round based on observed data. In the same section, we also described several realistic scenarios where observation selection under non-adaptive settings are more practical than the adaptive methods.
>
> On the other hand, 'adaptive-sampling PINN for inverse problem' as the reviewer described, has several key differences:
> 1. In inverse problems, the PINNs are used as inverse solvers, and the goal is to estimate unknown PDE parameters from given observed data. This is distinct from the ED problem as described above, which focuses on effectively optimizing the observation points for the inverse problem.
> 2. Sampling in this case usually refers to sampling methods for PINNs training, where most works focus on the selection of what are known as PDE residual or collocation points that enforces the PDE and IC/BC constraints in the PINN loss function [3], even when the PINNs are used for solving inverse problems. Instead, our work focuses on optimizing for the most informative observational data.
> 3. The only adaptive PINNs points selection work that considers the selection of observational points, as far as we are aware, is [4]. However, [4] focuses on the adaptive setting that involves choosing different observation points during the PINNs training process. Instead, our work focuses on the one-shot deployment setting where such frequent adjustments of sensor locations and observation points are impractical.
>
> [3] Wu et al. A comprehensive study of non-adaptive and residual-based adaptive sampling for physics-informed neural networks.
>
> [4] Lau et al. PINNACLE: PINN Adaptive ColLocation and Experimental points selection.
>
> ---
>
> > Can a trained PINN generalize to a new ED problem with new $\beta$? Can the model (including the meta-learning part) generalize to a new distribution of $\beta$?
>
> We would first like to clarify that the experimental design (ED) problem is defined over a given PDE system (Eq. 1 in the main paper) for a range of possible associated PDE parameter $\beta$ values, where the range is typically informed by domain knowledge. For the ED problem, we do not know what the true $\beta_0$ value is but want to find the most informative observation points (given budget constraint) such that we can best solve the inverse problem of estimating $\beta_0$.
>
> A PINN by itself, as mentioned earlier, cannot be used alone to tackle ED problems, but may be used as an inverse solver to estimate $\beta_0$. A trained PINN that was previously trained with another $\beta$ value could take in observation data and be re-trained to act as an inverse solver to estimate $\beta_0$ -- the original 'trained PINN' will essentially be the initialization for a newly trained PINN with the observation data.
>
> Regarding the selected observation points (i.e., ED solutions from the PIED methods) and whether they could be used on a new distribution of $\beta$, we would like to refer the reviewer to Fig. 11 in the Appendix, where we presented results for experiments where there is a mismatch in the distribution of the $\beta$ used during the ED stage and those during the testing stage. From the results we see that unlike other methods, PIED suffers little degradation in the predictive performances when the distribution of $\beta$ changes, suggesting that this generalization behavior is possible with PIED.
>
> ---
>
> Thank you once again for reviewing our work. We hope that our responses and clarifications will improve your evaluation of our paper.

---

> > ### Comment · Reviewer_CY8h · 2024-11-24
> >
> > Thank you for the detailed response. My main questions are addressed in the response, and I will raise my score to 6.

---

> ### Author Response · Authors · 2024-11-25
>
> Dear Reviewer CY8h, we would like to thank you once again for your review, and for improving your evaluation of our paper in response to our rebuttal. We are happy to provide further clarifications to address any other concerns that you may still have before the end of the rebuttal.

---

### Official Review · Reviewer_X3tF · 2024-11-03

**Soundness:** 3
**Presentation:** 3
**Contribution:** 3
**Rating:** 6
**Confidence:** 3

**Summary:**

The authors developed a physics-informed experimental design approach for the inverse problems to find the optimal and most informative design parameters. The paper is well-organized.

**Strengths:**

The theoretical analysis of PINN is thorough.

**Weaknesses:**

The effect of some experiments is not significant.

**Questions:**

1. In Fig. (c) and Fig. (d), the training loss and test error at final stage are similar for both methods. How can this be used to highlight the advantages of the proposed method.
2. How to define the value of μ in Eqs. (17) and (18). It is a very important parameter to determine the difficulty for solving the NS equations.
3. How to select the noise variance for different experiments. The noise variance may have a great influence for the prediction results. Can the method still be available when the noise variance is very large.

---

> ### Author Response · Authors · 2024-11-21
> **Response to Reviewer X3tF**
>
> We would like to thank the reviewer for taking the time to review our work, and we greatly appreciate the positive comments for our paper. We have provided clarifications to your questions below.
>
> ---
>
> > In Fig. (c) and Fig. (d), the training loss and test error at final stage are similar for both methods. How can this be used to highlight the advantages of the proposed method.
>
> We would like to highlight that Fig. 2c and 2d demonstrates how the meta-learned shared NN initialization leads to significantly faster convergence for the train loss and test error compared to random initialization -- for a given training step, the loss of PINNs using shared initialization is consistently lower, especially for earlier steps of the training.
>
> Such faster convergence provides computational cost advantages for our ED framework since less training steps are required to obtain relatively good approximations of the forward simulations and inverse solutions. Unlike inverse problems where the goal is to directly minimize the PINN loss for the PDE parameter, in the ED problem our goal is to find the most informative observation points which may not necessarily require the lowest possible loss. For example, in our FIST method, we only need to partially train the inverse PINNs to get sufficient learning signal to optimize for observation points through gradient-based methods, described in Sec. 4.2.
>
> We thank the reviewer for this comment, and we will look to add these extra discussion points to clarify the intentions and interpretations of the results in Figure 2.
>
> ---
>
> > How to define the value of $\mu$ in Eqs. (17) and (18). It is a very important parameter to determine the difficulty for solving the NS equations.
>
> In our experiments, the ground truth data is generated by fixing the constants $\mu$ and $\rho$ and then generating the data from fluid dynamics simulators accordingly. In the dataset we used, the value of $\mu$ is around $10^{-3}$, corresponding to a fluid with lower viscosity. We notice that we specified $\mu$ in our code (supplementary materials) but did not make this part clear in the description of the datasets used -- we will add them into the paper.
>
> ---
>
> > How to select the noise variance for different experiments. The noise variance may have a great influence for the prediction results. Can the method still be available when the noise variance is very large.
>
> In the real experiment cases (the groundwater and the cell experiments), we do not add additional noise into the experimental data, since they have already been collected from real experiments where there already exists actual noise, which can be seen in the visualization of those data in Figure 5. We find that these cases provide an illustration of the level of noise and even the structure of noise that are already experienced in realistic scientific settings (beyond simple Gaussian noise with fixed variance), and therefore use the data as is. In our main paper, we demonstrate that PIED is still able to perform well in these scenarios.
>
> In the synthetic experiments that we presented, we simulated more realistic experimental factors by injecting a small Gaussian noise to observation values and truncating the observation values to reduce its fidelity/simulate sensor limitations during the inference stage. These values are specified in our code (in the supplementary materials), but we will also add those that are missing into our Appendix.
>
> To demonstrate that **PIED still works even when there is larger noise variance** in the true observations for the synthetic experiments, we have run additional experiments on the Eikonal equation setting where the observation data has a higher noise level. We added Gaussian noise with SD 0.1 to the observation values and truncated them to the nearest 2 decimal places. This is a large value of noise since the readings in this case are in the order of at most $10^0$ or $10^1$. The results are as presented below. As we can see, all algorithms perform worse than before as expected due to increased noise, however PIED still is able to outperform all the other benchmarks.
>
> |             | Eikonal (Noisy) ($\times 10^{1}$)   |
> |:--------|:-----------------|
> | Random      | 3.63 (0.17)                         |
> | Grid        | 3.53 (0.06)                         |
> | MI          | 3.53 (0.16)                         |
> | VBOED       | 6.25 (1.23)                         |
> | FIST (ours) | **2.91 (0.51)**               |
> | MoTE (ours) | **2.88 (0.32)**                |
>
> ---
>
> Thank you once again for your comments. We hope our clarifications and responses have adequately addressed your concerns, and would improve your evaluation of our paper.

---

> ### Author Response · Authors · 2024-11-25
> **Gentle reminder for Reviewer X3tF**
>
> As we near the end of the discussion period, we would like to thank the reviewer again for their comments, and provide a gentle reminder that we have posted a response to your comments. May we please check if our responses have addressed your concerns and improved your evaluation of our paper?

---

> ### Author Response · Authors · 2024-12-03
> **Gentle reminder for Reviewer X3tF**
>
> As we are less than 8 hours to the end of the rebuttal, may we please check if our responses have addressed your concerns and improved your evaluation of our paper? We are happy to provide further clarifications to address any other concerns that you may still have before the end of the rebuttal.
>
> Thank you very much.

---

### Official Review · Reviewer_QnMu · 2024-11-03

**Soundness:** 3
**Presentation:** 3
**Contribution:** 3
**Rating:** 8
**Confidence:** 2

**Summary:**

This paper suggests PIED, a method for optimal experimental design for PDE inverse problems via Physics-informed neural networks (PINN). The PIED framework consists of three steps: 1. A PINN to learn a forward simulator that maps parameters $\beta$ to solutions of the PDE $u_\beta$. 2. An observation selector that returns noisy predicted observations $\tilde Y= u_\beta(X) + \epsilon$ at a fixed number of sensor placements $X$. 3. A PINN inverse solver that predicts the ground truth parameter $\hat \beta(X, \tilde Y)$ from $X$ and $\tilde Y$. 4. The optimal sensor placements $X$ are found by minimising the MSE between $\beta$ and $\hat \beta(X, \tilde Y)$ which requires back-propagation through the inverse problem solver. The paper claims the following contributions and innovations:
- The weight initialisations of all PINN-based components are shared and from a pretrained model which stabilises and accelerates training.
- The forward simulator for various choices of $\beta$ are learned in parallel
- The mean square error between $\hat\beta(X, \tilde Y)$ and $\beta$ is approximated by learning the inverse solver with a smaller number of training steps (FIST criterion for ED)
- The mean square error between $\hat\beta(X, \tilde Y)$ and $\beta$ is approximated through a linearisation of the PINN training dynamics (MoTE criterion for ED)

The framework is tested on optimal experimental design for sensor placement on Eikonal, Wave, and Navier-Stokes equations as well as groundwater and cell growth dynamics, and benchmarked against Bayesian experimental design, random sensor placements, and sensor placements on a grid.

**Strengths:**

- Overall, the paper is well written and seems methodologically sound.
- The paper addresses a challenging experimental design problem with little existing approaches.
- There is sufficient experimental results to support to proposed approach.
- The paper combines an interesting set of ideas:
	- Meta learning and shared weights for more efficient PINN training.
	- Approximate training dynamics through of the inverse solver to make back-propagation through the inverse solver feasible.

**Weaknesses:**

- The approach is mostly motivated by the comparison with classical simulators, but other types of neural surrogate models for the forward simulator are not mentioned. This is probably because most such approaches like Fouier Neural Operators don't investigate the ED downstream task. Nevertheless, this made me wonder how much this approach actually relies on Physics-informed neural networks. For example, the processing in parallel threads would be typical for all approaches that target ED with a neural network.
- The description of the algorithm/framework is a little inconsistent. The PIED framework in Figure 1 suggests that the PIED framework is trained end-to-end, while Algorithm 3 in the appendix clarifies that training proceeds in three phases that are not interleaved: Optimising for initial weights,  training to define ED criteria, optimisation of ED criterion. I wish that this was clearer from the main text.

**Questions:**

- I would suggest to describe the PIED framework as described in algorithm 3 more clearly in the main text of the paper
- After the experimental design is complete and sensor placements $X$ are obtained: Are the obtained sensor placements supposed to be universally informative and optimal for various downstream tasks of the practitioner, also those that involve classical simulators, or is the intention to always perform inference of the parameter $\beta$ with the trained inverse PINN (which may be inaccurate compared to a slower classical solver).
- Related to the previous question: Is it possible that the PINN memorises information about the PDE system at arbitrary evaluation points through the shared initialisation $\theta_{SI}$, such that the sensor locations $X$ are not the most informative to make predictions about the system, but only the most informative when we additionally have access to a PINN with parameters $\theta_{SI}$. In other words, could it be that the optimal experimental design is different when one wants to use classical simulators + MCMC to solve the inverse problem?

---

> ### Author Response · Authors · 2024-11-21
> **Response to Reviewer QnMu (Part 1/2)**
>
> We would like to thank the reviewer for taking the time to review our work, and we greatly appreciate the positive comments for our paper. We have provided clarifications to your questions below.
>
> ---
>
> > The approach is mostly motivated by the comparison with classical simulators, but other types of neural surrogate models for the forward simulator are not mentioned. This is probably because most such approaches like Fouier Neural Operators don't investigate the ED downstream task. Nevertheless, this made me wonder how much this approach actually relies on Physics-informed neural networks. For example, the processing in parallel threads would be typical for all approaches that target ED with a neural network.
>
> We would like to highlight that while one of the motivations of our work is the use of PINNs in lieu of classical simulators as forward simulators, another important advantage of using PINNs is that the same neural architecture could be used for **both the forward simulators and inverse solvers**. Our framework builds on this to provide significant performance advantages, as described in Lines 274-277 and described further in Sec. 4 of our main paper.
>
> Other neural surrogate models such as deeper neural operators (e.g., FNOs as the reviewer mentioned) can potential serve as powerful forward simulators with sufficient training data, but in practice these models will require very large amount of data that would themselves need to be generated from expensive forward simulations or experiments, making such models not as suitable for the data-constraint settings that we are considering. Physics-informed deep neural operators may be able to reduce training data requirements, but are more complex and challenging to train than PINNs and also do not readily work as inverse solvers like PINNs. However, we agree that this is a potential direction for future work, as we mentioned in the conclusion of our paper (Line 539).
>
> Nonetheless, to demonstrate the suitability of PINNs in our inverse problem setting, we investigate other NN-based ED method by running the neural likelihood estimation (NLE) method proposed by [1] as a benchmark, where a NN is used to estimate the likelihood of the observation outputs.
> In this sense, NLE is similar to FNOs in using a NN as a surrogate for the system dynamics, however is different in that the observation likelihood is learned instead of the observation output itself.
>
> We include some additional results for the NLE methods in the table below. From the table, we see that PIED is still able to outperform this baseline given the same forward simulation budget.
>
> |             | Eikonal ($\times 10^{1}$)   | Groundwater ($\times 10^{1}$)   | Cell Growth ($\times 10^{0}$)   |
> |:------------|:----------------------------|:--------------------------------|:--------------------------------|
> | NLE         | 1.63 (0.22)                 | 2.27 (1.59)                     | 3.65 (1.10)                     |
> | FIST (ours) | **0.74 (0.05)**        | **1.93 (0.27)**           | **2.62 (0.36)**           |
> | MoTE (ours) | **0.76 (0.04)**        | **2.00 (0.95)**            | 2.83 (0.27)                     |
>
> [1] Papamakarios et al. Sequential Neural Likelihood: Fast Likelihood-free Inference with Autoregressive Flows.
>
> ---
>
> > The PIED framework in Figure 1 suggests that the PIED framework is trained end-to-end, while Algorithm 3 in the appendix clarifies that training proceeds in three phases that are not interleaved.
>
> > I would suggest to describe the PIED framework as described in algorithm 3 more clearly in the main text of the paper
>
> We thank the reviewer for pointing out that Fig. 1b may give the impression that the PIED framework is trained end-to-end -- we will adjust the paper to describe Alg. 3 more explicitly in the main paper as suggested. We had originally intended for Fig. 1b in Sec.3 to mainly provide an overview of the different components in our PIED framework, with a comparison on how it mirrors the components required for solving the inverse problem during inference time on real systems (Fig. 1a). We then built on this by describing the training and optimization methods used in the implementation of PIED later in Sec. 4, with details including an overview and Alg. 3 in Appendix G. However, we agree that our visualization could be more explicit in describing the complete PIED framework implementation in the main paper, and will attempt to incorporate these points into the paper.

---

> ### Author Response · Authors · 2024-11-21
> **Response to Reviewer QnMu (Part 2/2)**
>
> > Are the obtained sensor placements supposed to be universally informative and optimal for various downstream tasks of the practitioner, also those that involve classical simulators...
>
> > ...could it be that the optimal experimental design is different when one wants to use classical simulators + MCMC to solve the inverse problem?
>
> Our PIED framework is primarily designed to be used with PINNs as inverse solvers, which allows for significant computational advantage including amortized training from the PIED implementation process. We would like to highlight that as we are solving inverse problems, which involves estimating PDE parameters $\beta$, the estimated parameters can be directly applied to classical simulators for downstream forward prediction tasks if desired, which as the reviewer pointed out may in some cases provide more accurate predictions compared to PINNs.
>
> Nevertheless, the optimal points selected by PIED for PINNs inverse solvers would likely have common desired characteristics for good sensor points for inverse problems in general, as it can be intuitively interpreted as optimizing for points that can best distinguish among different reference $\beta$ values. To see this, note that the observation selector component (Fig. 1) can be interpreted as an information bottleneck selecting the most informative observation points from output of forward simulators such that the inverse solvers can distinguish and identify the various different $\beta$ values.
>
> We demonstrate this empirically by observing how the inverse solvers which uses PINNs and which uses classical simulators perform when fed with observations which are selected by PIED. From the table below, we see that even when the inverse problem is solved using classical simulators, the observations selected by PIED are still able to accurately recover the correct inverse parameter and improve from the baseline.
>
> |               | Groundwater ($\times 10^{1}$)   |
> |:--------------|:--------------------------------|
> | Random (Sim)  | 2.74 (1.95)                     |
> | Random (PINN) | 3.44 (2.71)                     |
> | FIST (Sim)    | 2.51 (0.75)                     |
> | FIST (PINN)   | 1.93 (0.27)            |
>
> ---
>
> > Is it possible that the PINN memorises information about the PDE system at arbitrary evaluation points through the shared initialisation $\theta_{SI}$, such that the sensor locations $X$ are not the most informative to make predictions about the system, but only the most informative when we additionally have access to a PINN with parameters $\theta_{SI}$.
>
> We would like to highlight that as the shared initialization $\theta_{SI}$ is meta-learned over many different reference $\beta$ values, it is actually less likely to memorize information about a specific PDE system at arbitrary evaluation points but instead **learn broad, common features** across different reference $\beta$ values instead, as visualized in Fig. 2b and Fig. 3b. This helps to speed up the PINNs training process, as can be seen in Fig. 2c-d and Fig. 10. In practice, it would be beneficial to use an inverse solver PINN initialized with $\theta_{SI}$ to make predictions on the true system $\beta$ value since these broad, common features would not have to be relearned.
>
> Nevertheless, we have run additional experiments (results below) comparing how PIED performs when using and when not using a shared parameter initialization for the ED process and during the inference stage.
> From the results, we see that when not using $\theta_{SI}$, we are still able to achieve good performances.
>
> |             | Groundwater ($\times 10^{1}$)   |
> |:------------|:--------------------------------|
> | Random      | 3.44 (2.71)                     |
> | FIST w/o SI | 2.66 (3.01)                     |
> | FIST w/ SI  | 1.93 (0.27)            |
> | MoTE w/o SI | 2.17 (4.49)                     |
> | MoTE w/ SI  | 2.00 (0.95)            |
>
> ---
>
> Thank you again for your insightful comments. We hope that we have sufficiently addressed your queries, and that our response will help strengthen your support for our paper.

---

> ### Author Response · Authors · 2024-11-25
> **Gentle reminder for Reviewer QnMu**
>
> As we near the end of the discussion period, we would like to thank the reviewer again for their comments, and provide a gentle reminder that we have posted a response to your comments. May we please check if our responses have addressed your concerns and improved your evaluation of our paper?

---

> > ### Comment · Reviewer_QnMu · 2024-12-01
> >
> > I thank the authors for their detailed response to my questions, including additional experimental results to support their points. My concerns were addressed and the responses improved my understanding of the benefits of PINN-based experimental design. In addition, experimental design is a field that demands innovations to overcome the often prohibitive cost of classical approaches. I therefore would like to increase my score to an accept.

---

> ### Author Response · Authors · 2024-12-03
>
> Dear Reviewer QnMu, we would like to thank you once again for your review, and for improving your evaluation of our paper in response to our rebuttal.

---

### Official Review · Reviewer_XCZE · 2024-11-06

**Soundness:** 3
**Presentation:** 3
**Contribution:** 3
**Rating:** 8
**Confidence:** 2

**Summary:**

The paper proposes a novel algorithm for solving the inverse problem in PDEs - that is, estimating the parameters of a PDE governing the dynamic characteristics of a system.  In particular, the paper assumes that (a) obtaining (x,y) observations of the system is expensive, so we must select our observation points carefully; and (b) observations require an initial setup that we cannot (practically) repeat, so we must specify our test points up-front and not dynamically as in e.g. Bayesian Optimization.

To tackle this problem the paper suggests physics-informed experimental design (PIED) that uses two sets of PINNs to select appropriate test points for a given system of PDEs (with a-priori unknown parameters).  The general approach uses a set of PINNs operating as forward simulators to generate functions satisfying the PDEs, sampling these for a set of points X, then using PINNs as inverse solvers to estimate the PDE parameters from the observations.  The efficacy of the points X is measured as the difference between the "real" PDE parameters (used in the forward simulators) and the corresponding estimated estimated PDE parameters.

**Strengths:**

The algorithm is certainly interesting.  The PIED framework certainly looks practical.  The motivation behind and justification of each step is presented, and the experimental results look good.

**Weaknesses:**

One point that needs to be address in the paper is that of computational cost.  After all, one of the motivations of using PINNs in parallel is computational efficiency, so it would be good to have a comparison in terms of same.

**Questions:**

See previous.

---

> ### Author Response · Authors · 2024-11-21
> **Response to Reviewer XCZE**
>
> We would like to thank the reviewer for taking the time to review our work, and we greatly appreciate the positive comments for our paper. We have provided clarifications to your question below.
>
> ---
>
> > One point that needs to be address in the paper is that of computational cost. After all, one of the motivations of using PINNs in parallel is computational efficiency, so it would be good to have a comparison in terms of same.
>
> We thank the reviewer for the comment regarding comparisons on computational cost. We would like to refer the reviewer to Figure 6 and Lines 510-530 of the main paper, where we compared both the errors and computational costs of the (1) benchmark methods using numerical solvers, (2) benchmark methods adapted to use PINNs for forward simulation and solving inverse problem, and (3) our methods using PINNs, for the 2D Eikonal scenario. Notice that all benchmark methods using numerical solvers had significantly higher computational costs and worse errors compared to the benchmarks adapted to use PINNs and our methods, demonstrating the advantages of using PINNs in parallel for computational efficiency as the reviewer had commented on.
>
> In all of our experiments, we also compared our proposed methods which uses PINNs, against the higher bar of benchmark methods adapted to use PINNs, which also benefit from the advantages of using PINNs. As can be seen in our various empirical results, our proposed methods consistently outperforms these benchmarks, validating our algorithm design choice of PIED using parallel PINNs and employing optimization that exploits their training dynamics.

---

> ### Author Response · Authors · 2024-11-25
> **Gentle reminder for Reviewer XCZE**
>
> As we near the end of the discussion period, we would like to thank the reviewers again for their comments, and provide a gentle reminder that we have posted a response to your comments. We hope that our responses have addressed your concerns and strengthened your support for our paper.

---

> ### Author Response · Authors · 2024-12-03
> **Gentle reminder for Reviewer XCZE**
>
> As the rebuttal is coming to a close, we would like to thank you once again for your review of our paper, and hope that we have provided adequate clarifications to your questions. We would be happy to provide further responses otherwise.

---

### Meta-Review · Area_Chair_89uX · 2024-12-22

**Metareview:**

This paper proposes a new method for optimal experimental design for PDE inverse problems via Physics-informed neural networks (PINN).  The approach entails learning a forward simulator that maps parameters to PDE solutions;  an observation selector that returns noisy predicted observations  at a fixed number of sensor placements; and an inverse solver that predicts the ground truth parameter. This sets up an objective function for finding optimal sensor placements.  Favorable experimental comparisons against Bayesian experimental design, random sensor placements, and sensor placements on a grid for Eikonal, Wave, and Navier-Stokes equations as well as groundwater and cell growth dynamics is the primary strength of this paper. Computational cost, novelty and some remarks on empirical significance were brought up as weaknesses, but the paper received unanimously positive reviews.

**Additional Comments On Reviewer Discussion:**

For computational cost concerns: the authors included results to show that other baselines have significantly higher computational costs and worse errors compared to the benchmarks adapted to their methods, demonstrating advantages of their parallel implementation.

Can the method still work when the noise variance is very large: Again, the authors provided additional experiments  on the Eikonal equation setting where the observation data has a higher noise level, showing favorable results.

Additional comparisons were also included against NN-based experiment design methods.

---

### Decision · Program_Chairs · 2025-01-22

Accept (Poster)